# What Can We Learn from State Space Models for Machine Learning on Graphs?

## Abstract

Machine learning on graphs has recently found extensive applications across domains. However, the commonly used Message Passing Neural Networks (MPNNs) suffer from limited expressive power and struggle to capture long-range dependencies. Graph transformers offer a strong alternative due to their global attention mechanism, but they come with great computational overheads, especially for large graphs. In recent years, State Space Models (SSMs) have emerged as a compelling approach to replace full attention in transformers to model sequential data. It blends the strengths of RNNs and CNNs, offering a) efficient computation, b) the ability to capture long-range dependencies, and c) good generalization across sequences of various lengths. However, extending SSMs to graph-structured data presents unique challenges due to the lack of canonical node ordering in graphs. In this work, we propose Graph State Space Convolution (GSSC) as a principled extension of SSMs to graph-structured data. By leveraging global permutation-equivariant set aggregation and factorizable graph kernels that rely on relative node distances as the convolution kernels, GSSC preserves all three advantages of SSMs. We demonstrate the provably stronger expressiveness of GSSC than MPNNs in counting graph substructures and show its effectiveness across 11 real-world, widely used benchmark datasets. GSSC achieves the best results on 6 out of 11 datasets with all significant improvements compared to the state-of-the-art baselines and second-best results on the other 5 datasets. Our findings highlight the potential of GSSC as a powerful and scalable model for graph machine learning. Our code is available at `https://anonymous.4open.science/r/GSSC-5ED8`.

## 1 Introduction

Machine learning for graph-structured data has numerous applications in molecular graphs (Duvenaud et al., 2015; Wang et al., 2021), drug discovery (Xiong et al., 2021; Stokes et al., 2020), and social networks (Fan et al., 2019; Guo & Wang, 2020). In recent years, Message Passing Neural Networks (MPNNs) have been arguably the most popular neural architecture for graphs (Kipf & Welling, 2016; Fung et al., 2021; Veličković et al., 2018; Xu et al., 2018; Corso et al., 2020; Zhou et al., 2020), but they also suffer from many limitations, including restricted expressive power (Xu et al., 2018; Morris et al., 2019), over-squashing (Di Giovanni et al., 2023; Topping et al., 2022; Nguyen et al., 2023), and over-smoothing (Rusch et al., 2023; Chen et al., 2020a; Keriven, 2022). These limitations could harm the models' performance. For example, MPNNs cannot capture long-range dependencies (Dwivedi et al., 2022) or detect subgraphs like cycles that are important in forming ring systems of molecular graphs (Chen et al., 2020b).

Adapted from the vanilla transformer in sequence modeling (Vaswani et al., 2017), graph transformers have attracted growing research interests because they may alleviate these fundamental limitations of MPNNs (Kreuzer et al., 2021; Kim et al., 2022; Rampášek et al., 2022; Chen et al., 2022a; Dwivedi & Bresson, 2020). By attending to all nodes in the graph, graph transformers are inherently able to capture long-range dependencies. However, the global attention mechanism ignores graph structures and thus requires incorporating positional encodings (PEs) of nodes (Rampášek et al., 2022) that encode graph structural information. For example, the information of relative distance between nodes has been leveraged in attention computation (Li et al., 2020; Wang et al., 2022; Ying et al., 2021). Moreover, the full attention computation scales quadratically in terms of the length of the sequence or the number of nodes in the graph. This computational challenge motivates the study of linear-time

transformers by incorporating techniques such as low-rank (Katharopoulos et al., 2020; Wang et al., 2020; Child et al., 2019; Choromanski et al., 2020; Yang et al., 2023), sparse approximations (Indyk & Motwani, 1998; Kitaev et al., 2020; Daras et al., 2020; Zandieh et al., 2023; Han et al., 2023), or Taylor expansion (Arora et al., 2024) of the attention matrix. Some specific designs of scalable transformers for large-scale graphs are also proposed (Wu et al., 2022; Chen et al., 2022b; Shirzad et al., 2023; Wu et al., 2023; Kong et al., 2023; Wu et al., 2024). Nevertheless, none of these variants have been proven to be consistently effective across different domains (Miao et al., 2024).

State Space Models (SSMs) (Gu et al., 2021a;b; Gu & Dao, 2023) have recently demonstrated promising potentials for sequence modeling. Adapted from the classic state space model (Kalman, 1960), SSMs can be seen as a hybrid of recurrent neural networks (RNNs) and convolutional neural networks (CNNs). It is a temporal convolution that preserves translation invariance and thus allows good generalization to sequences longer than those used for training. Meanwhile, this class of models has been shown to capture *long-range dependencies* both theoretically and empirically (Gu et al., 2020; Tay et al., 2020; Gu et al., 2021b;a). Finally, it can be efficiently computed in *linear or near-linear time* via the recurrence mode or the parallelizable operations. These advantages make the SSM a strong candidate as an alternative to transformers (Mehta et al., 2022; Ma et al., 2022; Fu et al., 2022; Wang et al., 2023; Sun et al., 2024).

Given the great potential of SSMs, there is increasing interest in generalizing them for graphs as an alternative to graph transformers (Wang et al., 2024a; Behrouz & Hashemi, 2024). The main technical challenge is that SSMs are defined on sequences that are ordered and causal, i.e., have a linear structure. Yet, graphs have complex topology, and no canonical node ordering can be found. Naive tokenization (e.g., sorting nodes into a sequence in some ways) breaks the inductive bias - permutation symmetry - of graphs, which consequently cannot faithfully represent graph topology, and may suffer from poor generalization.

In this study, we go back to the fundamental question of how to build SSMs for graph data. Instead of simply tokenizing graphs and directly applying existing SSMs for sequences (which may break the symmetry), we argue that principled graph SSMs should inherit the advantages of SSMs in capturing long-range dependencies and being efficient. Simultaneously, they should also preserve the permutation symmetry of graphs to achieve good generalization. With this goal, in this work:

- We identify that the key components enabling SSMs for sequences to be long-range, efficient, and well-generalized to longer sequences, is the use of a global, factorizable, and translation-invariant kernel that depends on relative distances between tokens. This relative-distance kernel can be factorized into the product of absolute positions, crucial to achieving linear-time complexity.

- This observation motivates us to design Graph State Space Convolution (GSSC) in the following way: (1) it leverages a global *permutation equivariant set aggregation* that incorporates all nodes in the graph; (2) the aggregation weights of set elements rely on relative distances between nodes on the graph, which can be factorized into the "absolute positions" of the corresponding nodes, i.e., the PEs of nodes. By design, the resulting GSSC is inherently permutation equivariant, long-range, and linear-time. Besides, we also demonstrate that GSSC is more powerful than MPNNs and can provably count at least 4-paths and 4-cycles.

- Empirically, our experiments demonstrate the high expressivity of GSSC via graph substructure counting and validate its capability of capturing long-range dependencies on Long Range Graph Benchmark (Dwivedi et al., 2022). Results on 10 real-world, widely used graph machine learning benchmark datasets (Hu et al., 2020; Dwivedi et al., 2023; 2022) also show the consistently superior performance of GSSC, where GSSC achieves best results on 7 out of 10 datasets with all significant improvements compared to the state-of-the-art baselines and second-best results on the other 3 datasets. Moreover, it has much better scalability than the standard graph transformers in terms of training and inference time.

## 2 PRELIMINARIES

**Graphs and Graph Laplacian.** Let $\mathcal{G} = (\mathcal{V}, \mathcal{E})$ be a undirected graph, where $\mathcal{V}$ is the node set and $\mathcal{E}$ is the edge set. Suppose $\mathcal{G}$ has $n$ nodes. Let $\boldsymbol{A} \in \mathbb{R}^{n \times n}$ be the adjacency matrix of $\mathcal{G}$ and $\boldsymbol{D} = \text{diag}([\sum_j \boldsymbol{A}_{1,j}, ..., \sum_j \boldsymbol{A}_{n,j}])$ be the diagonal degree matrix. The (normalized) graph Laplacian is defined by $\boldsymbol{L} = \boldsymbol{I} - \boldsymbol{D}^{-1/2} \boldsymbol{A} \boldsymbol{D}^{-1/2}$ where $\boldsymbol{I}$ is the $n$ by $n$ identity matrix.

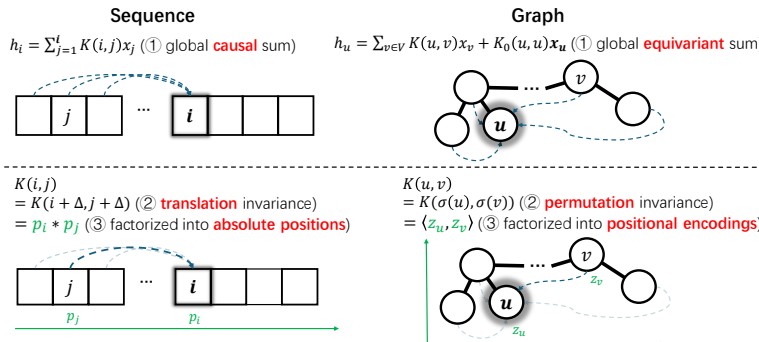

Figure 1: Comparison of Sequence State Space Conv. (left) and Graph State Space Conv. (right).

**State Space Models.** State space model is a continuous system that maps a input function $x(t)$ to output $h(t)$ by the following first-order differential equation: $\frac{d}{dt}h(t) = Ah(t) + Bx(t)$. This system can be discretized by applying a discretization rule (e.g., bilinear method (Tustin, 1947), zero-order hold (Gu & Dao, 2023)) with time step $\Delta$. Suppose $h_i := h(i \cdot \Delta)$ and $x_i := x(i \cdot \Delta)$. The discrete state space model becomes a recurrence process:

$$h_i = \bar{A}h_{i-1} + \bar{B}x_i, \tag{1}$$

where $\bar{A} = f_A(\Delta, A)$ and $\bar{B} = f_B(\Delta, B)$ depend on the specific discretization rule. The recurrence Eq. (1) can be computed equivalently by a global convolution:

$$h_i = \sum_{j=1}^{i} \bar{A}^{i-j}\bar{B}x_j. \tag{2}$$

In the remaining of this paper, we call convolution Eq. (2) **state space convolution (SSC)**. Notably, the state space model enjoys three key advantages simultaneously:

- *Translation equivariance.* A translation to input $x_i \rightarrow x_{i+\delta}$ yields the same translation to output $h_i \rightarrow h_{i+\delta}$.

- *Long-range dependencies.* The feature $h_i$ of $i$-th token depends on all preceding tokens' features $x_1, x_2, ..., x_{i-1}$. With a properly chosen structured matrix $\bar{A}$, e.g., HiPPO (Gu et al., 2020), low-rank (Gu et al., 2021a) or diagonal matrices (Gupta et al., 2022), the gradient norm $\|\partial h_i/\partial x_{i-j}\|$ does not decay as $j$ goes large. This is different to the fixed-size receptive field in CNNs (LeCun et al., 1998; Krizhevsky et al., 2012) and the vanishing gradient norm in RNNs (Pascanu, 2013).

- *Computational efficiency and parallelism.* To computing $h_1, h_2, ..., h_n$, it adopts either recurrence (Eq. 1) or convolution (Eq. 2). Near-linear-time algorithms are introduced to provide parallelism for efficient training, e.g., FFT (Gu et al., 2021a; Karami & Ghodsi, 2024), block-decomposition matrix multiplication (Dao & Gu, 2024) for convolution, and parallel scan (Gu & Dao, 2023) for recurrence.

**Notation.** Suppose $x, y$ are two vectors of dimension $n$. Denote $\langle x, y \rangle = \sum_{i=1}^{n} x_i y_i$ as the inner product, $x \odot y = (x_1 y_1, x_2 y_2, ...)$ be the element-wise product. We generally denote the hidden dimension by $m$, and the dimension of positional encodings by $d$.

## 3 GRAPH STATE SPACE CONVOLUTION

### 3.1 GENERALIZING STATE SPACE CONVOLUTION TO GRAPHS

Like standard SSC (Eq. 2), the desired graph SSC should keep the good capturing of long-range dependencies as well as linear-time complexity and parallelizability. Meanwhile, permutation equivariance as a strong inductive bias of graph-structured data should be preserved by the model as well to improve generalization.

Our key observations of SSC start with the fact that the convolution kernel $\bar{A}^{i-j}$ encoding the relative distance $i - j$ between token $i$ and $j$ allows for a natural factorization:

$$h_i = \sum_{j=1}^{i} \bar{A}^{i-j}\bar{B}x_j = \bar{A}^i \sum_{j=1}^{i} \bar{A}^{-j}\bar{B}x_j. \tag{3}$$

Generally, $\sum_j K(i,j)x_j$ with a generic kernel $K(i,j)$ requires quadratic-time computation and may not capture translation invariant patterns for variable-length generalization. For SSC (Eq. 3), however, it captures translation invariant patterns by adopting a **translation-invariant** kernel $K(i-j)$ that only depends on the relative distance $i-j$, which gives the generalization power to sequences even longer than those used for training (Giles & Maxwell, 1987; Kazemnejad et al., 2024). Moreover, SSC attains computational efficiency by leveraging the **factorability** of its particular choice of the relative distance kernel $K(i-j) = \bar{A}^{i-j} = \bar{A}^i \cdot \bar{A}^{-j}$, where the factors only depend on the *absolute positions* of token $i$ and $j$ respectively. To compute $h$, one construct $[\sum_{j=1}^{1} \bar{A}^{-j} \bar{B} x_j, ..., \sum_{j=1}^{n} \bar{A}^{-j} \bar{B} x_j]$ using prefix sum with complexity $O(n)$, and then readout $h_i$ by multiplying $\bar{A}^i$ for all $i = 1, 2, ..., n$ in parallel with complexity $O(n)$. Finally, the **global causal sum** $\sum_{j \leq i}$ helps capture global dependencies. Overall, the advantages of SSC are attributed to these three aspects.

Inspired by these insights, a natural generalization of SSC to graphs, written generally as $h_v = \sum_{u \in V} K(v,u)x_u$, is expected to adopt a **permutation-invariant** kernel $K(v,u)$, $v, u \in \mathcal{V}$ to capture the inductive bias of graphs. The kernel should be **factorizable** $K(v,u) = z_v^\top z_u$ with certain notion of node absolute position for computational efficiency, and the convolution should perform **global pooling** across the entire graph to capture global dependencies. Note that causal sum $\sum_{j \leq i}$ is replaced by global pooling $\sum_{u \in V}$ due to the lack of causality in the order of nodes.

Fortunately, a systematic strategy can be adopted to design such kernels. First, the kernel $K(v,u)$ should depend on some notions of relative distance between nodes to capture graph topology and preserve permutation invariance. The choices include but are not limited to shortest-path distance, random walk landing probability (Li et al., 2019) (such as PageRank (Page et al., 1999)), heat (diffusion) distance (Chung, 2007), resistance distance (Xiao & Gutman, 2003; Palacios, 2001), etc. Many of them have been widely adopted as edge features for existing models, e.g., GNNs (You et al., 2021; Li et al., 2020; Zhang & Li, 2021; Chien et al., 2021; Velingker et al., 2024; Nikolentzos & Vazirgiannis, 2020) and graph transformers (Kreuzer et al., 2021; Rampášek et al., 2022; Ma et al., 2023; Mialon et al., 2021). More importantly, all these kernels can be factorized into some weighted inner product of Laplacian eigenvectors (Belkin & Niyogi, 2003). Here, Laplacian eigenvectors, also known as Laplacian positional encodings (LPE) (Wang et al., 2022; Dwivedi et al., 2023; Lim et al., 2022), play the role of the absolute positions of nodes in the graph. Formally, consider the eigendecomposition $\boldsymbol{L} = \boldsymbol{V}\boldsymbol{\Lambda}\boldsymbol{V}^\top$ and let $p_u = [\boldsymbol{V}_{u,:}]^\top$ be the LPE for node $u$. Then a relative-distance kernel $K(u,v)$ can be generally factorized into $K(u,v) = p_u^\top(\phi(\boldsymbol{\Lambda}) \odot p_v)$ for certain functions $\phi$. For instance, diffusion kernel satisfies $[\phi(\boldsymbol{\Lambda})]_k = \exp(-t\lambda_k)$ for some time parameter $t$.

**Graph State Space Convolution (GSSC).** Given the above observations, we are ready to present GSSC as follows. Given the input node features $x_u \in \mathbb{R}^m$, the $d$-dim Laplacian positional encodings $p_u = [\boldsymbol{V}_{u,1:d}]^\top \in \mathbb{R}^d$, and the corresponding $d$ eigenvalues $\boldsymbol{\Lambda}_d = [\lambda_1, ..., \lambda_d]^\top$, the output node representations $h_u \in \mathbb{R}^m$ follow

$$h_u = \sum_{v \in \mathcal{V}} \langle z_u \boldsymbol{W}_q, z_v \boldsymbol{W}_k \rangle \odot \boldsymbol{W}_o x_v + \langle z_u \boldsymbol{W}_{sq}, z_u \boldsymbol{W}_{sk} \rangle \odot \boldsymbol{W}_s x_u, \tag{4}$$

$$= \langle z_u \boldsymbol{W}_q, \sum_{v \in \mathcal{V}} z_v \boldsymbol{W}_k \odot \boldsymbol{W}_o x_v \rangle + \langle z_u \boldsymbol{W}_{sq}, z_u \boldsymbol{W}_{sk} \odot \boldsymbol{W}_s x_u \rangle. \tag{5}$$

Here $z_u = [\phi_1(\boldsymbol{\Lambda}_d) \odot p_u, ..., \phi_m(\boldsymbol{\Lambda}_d) \odot p_u] \in \mathbb{R}^{d \times m}$ represents the eigenvalue-augmented PEs from raw $d$-dim PEs $p_u$ and $\phi_\ell : \mathbb{R}^d \to \mathbb{R}^d$ are learnable permutation equivariant functions w.r.t. $d$-dim axis (i.e., equivariant to permutation of eigenvalues). All $\boldsymbol{W} \in \mathbb{R}^{m \times m}$ with different subscripts are learnable weight matrices. The inner product $\langle z_u \boldsymbol{W}_q, z_v \boldsymbol{W}_k \rangle \in \mathbb{R}^m$ only sums over the first $d$-dim axis. The term $z_u \boldsymbol{W}_{sk} \odot \boldsymbol{W}_s x_u$ in Eq.5 should be interpreted as first broadcasting $\boldsymbol{W}_s x_u$ from $\mathbb{R}^m$ to $\mathbb{R}^{d \times m}$ and then performing element-wise products. Note that GSSC is a generalization of SSC (Eq. 3) in the sense that:

- *Absolute position*: absolute position $\bar{A}^i$ is replaced by positional encodings $z_u$;

- *factorizable kernel*: the kernel $\bar{A}^i \bar{A}^{-j}$ in terms of the product of absolute positions is replaced by the inner product of graph positional encodings $\langle z_u \boldsymbol{W}_q, z_v \boldsymbol{W}_k \rangle$.

- *global sum*: casual sum $\sum_{j=1}^{i} \bar{A}^{-j} \bar{B} x_j$ is replaced by an equivariant global pooling. As graphs are not causal, which means a node $u$ only distinguishes itself (node $u$) from other nodes, we

Figure 2: Illustration of Graph State Space Convolution (GSSC).

adopt permutation equivariant global pooling consists of the term $z_u \boldsymbol{W}_{sk} \odot \boldsymbol{W}_s x_u$ denoting node $u$ itself, and $(\sum_{v \in \mathcal{V}} z_v \boldsymbol{W}_k) \odot \boldsymbol{W}_o x_v$ denoting all nodes in the graph.

Proposition 3.1 suggests that GSSC with learnable $\phi$ can capture long-range dependencies.

**Proposition 3.1.** *There exists $\phi$ such that for GSSC Eq. 4, the gradient norm $\|\partial h_u / \partial x_v\|$ does not decay as $spd(u, v)$ grows, where $spd$ denotes the shortest path distance.*

To sum up, the design of GSSC leads to the following key properties as desired.

**Remark 3.1.** *GSSC (Eq. 4) is (1) permutation equivariant: a permutation of node indices reorders $h_u$ correspondingly; (2) long-range: as suggested by Proposition 3.1; (3) linear-time: the complexity of computing $h_1, ..., h_n$ from $x_1, ..., x_n$ is $\mathcal{O}(nmd)$, where $n$ is the number of nodes and $m, d$ are hidden and positional encoding dimension; (4) stable: perturbation to graph Laplacian yields a controllable change of GSSC model output, because permutation equivariance and smoothness of $\phi_\ell$ ensure the stability of inner product $\langle z_u \boldsymbol{W}_q, z_v \boldsymbol{W}_k \rangle$ and model output, as shown in Wang et al. (2022); Huang et al. (2024). Stability is an enhanced concept of permutation equivariance and is crucial for out-of-distribution generalization (Huang et al., 2024).*

## 3.2 EXTENSIONS AND DISCUSSIONS

**Incorporating Edge Features.** SSMs (and the proposed GSSC) do not have a natural way to incorporate token-pairwise (edge) features. To address this inherent limitation, SSMs (and their graph extension GSSC) need to be paired with modules that can incorporate token-pairwise (edge) features, such as MPNNs adopted in previous graph Mambas (Wang et al., 2024a; Behrouz & Hashemi, 2024) and graph transformers (Rampášek et al., 2022; Chen et al., 2022a). SSMs and MPNNs complement either side by capturing global dependence via SSMs and edge features via MPNNs. This can be validated by the significant performance boost compared with using either module alone in practice.

**Selection Mechanism in SSMs.** It is known that SSMs lack of selection mechanism, i.e., the kernel $\bar{A}^{i-j}$ is only a function of positions $i, j$ and does not rely on the feature of tokens (Gu & Dao, 2023). To improve the content-aware ability of SSMs, Gu & Dao (Gu & Dao, 2023) proposed to make coefficients $\bar{A}, \bar{B}$ in Eq.1 data-dependent, i.e., replacing $\bar{A}$ by $\bar{A}_i := \bar{A}(x_i)$ and $\bar{B}$ by $\bar{B}_i := \bar{B}(x_i)$. This leads to a data-dependent convolution: $h_i = \sum_{j=1}^i \bar{A}_{i-1}\bar{A}_{i-2}...\bar{A}_j \bar{B}_j x_j$. Again, this convolution can be factorized into $h_i = \tilde{A}_i(\sum_{j=1}^i \tilde{A}_j^{-1} \bar{B}_j x_j)$, where $\tilde{A}_i := \bar{A}_{i-1}\bar{A}_{i-2}...\bar{A}_1$ can be interpreted as a data-dependent absolute position of token $i$, depending on features and positions of all preceding tokens. We can generalize this "data-dependent position" idea to GSSC, defining a data-dependent positional encodings $\tilde{z}$ as follows:

$$\tilde{z}_u = \sum_{v \in \mathcal{V}} \langle z_u \boldsymbol{W}_{dq}, z_v \boldsymbol{W}_{dk} \rangle (z_v \odot x_v) \boldsymbol{W}_{dv}, \tag{6}$$

where $z_u \odot x_u$ should be interpreted as first broadcasting $x_u$ from $\mathbb{R}^m$ to $\mathbb{R}^{r \times m}$ and then doing the element-wise product with $z_u$. Note that the new positional encodings $\tilde{z}_u$ rely on the features and positional encodings of all nodes, which reflects permutation equivariance. To achieve a selection mechanism, we can replace every $z_u$ in Eq. 4 by $\tilde{z}_u$. Thanks to the factorizable kernel $\langle z_u \boldsymbol{W}_{dq}, z_v \boldsymbol{W}_{dk} \rangle$, computing $\tilde{z}_u$ can still be done in $\mathcal{O}(nmd^2)$, linear w.r.t. graph size.

**Compared to Graph Spectral Convolution (GSC).** GSSC may also look similar to graph spectral convolution, which is usually in the form of $\boldsymbol{H} = \boldsymbol{V}\psi(\boldsymbol{\Lambda})\boldsymbol{V}^\top \boldsymbol{X}\boldsymbol{W}$, where $\boldsymbol{H} = [h_1^\top; ...; h_n^\top]^\top \in \mathbb{R}^{n\times m}$ and $\boldsymbol{X} = [x_1^\top; ...; x_n^\top]^\top \in \mathbb{R}^{n\times m}$ are a row-wise concatenation of features, $\psi(\boldsymbol{\Lambda}) = \text{diag}([\psi(\lambda_1), ..., \psi(\lambda_n)])$ is the spectrum-domain filtering with $\psi : \mathbb{R} \to \mathbb{R}$ being an element-wise function. Equivalently, it can be written as $h_u = \sum_{v\in\mathcal{V}}\langle p_u, \psi(\boldsymbol{\Lambda}) \odot p_v\rangle \boldsymbol{W}x_v$. There are several differences between GSSC (Eq. 4) and graph spectral convolution: (1) $\psi : \mathbb{R} \to \mathbb{R}$ is an element-wise filters, while $\phi : \mathbb{R}^m \to \mathbb{R}^m$ is a general permutation equivariant function that considers the mutual interactions between frequencies; (2) GSSC distinguishes a node itself (weight $\boldsymbol{W}_{sq}, \boldsymbol{W}_{sk}, \boldsymbol{W}_{so}$) and other nodes (weight $\boldsymbol{W}_q, \boldsymbol{W}_k, \boldsymbol{W}_o$), while GSC treats all nodes using the same weight $\boldsymbol{W}$. The former turns out to be helpful to express the diagonal element extraction ($\text{diag}(\boldsymbol{A}^k)$) in cycle counting. In comparison, GSC with non-distinguishable node features cannot be more powerful than 1-WL test (Wang & Zhang, 2022), while GSSC, is more powerful as shown later.

**Super-linear Computation of Laplacian Eigendecomposition.** GSSC requires the computation of Laplacian eigendecomposition as preprocessing. Although finding all eigenvectors and eigenvalues can be costly, (1) in real experiments, such preprocessing can be done efficiently, which may only occupy less than 10% wall-clock time of the entire training process (see Sec. 5.3 for quantitative results on real-world datasets); (2) we only need top-$d$ eigenvectors and eigenvalues, which can be efficiently found by Lanczos methods (Paige, 1972; Lanczos, 1950) with complexity $O(Ed)$ ($E$ is the number of edges) or similarly by LOBPCG methods (Knyazev, 2001); (3) one can also adopt random Fourier feature-based approaches to fast approximate those kernels' factorization (Smola & Kondor, 2003; Choromanski, 2023; Reid et al., 2024) without precisely computing the eigenvectors.

### 3.3 EXPRESSIVE POWER

We measure the expressivity of GSSC via graph distinguishing ability compared to WL test hierarchy.

**Proposition 3.2.** *GSSC is strictly more powerful than WL test and not more powerful than 3-WL test.*

We can also characterize the expressivity of GSSC via the ability to count graph substructures. Proposition 3.3 states that GSSC can count at least 3-paths and 3-cycles, which is strictly stronger than the counting power of MPNNs and GSC (both cannot count cycles (Chen et al., 2020b; Wang & Zhang, 2022)). Furthermore, if we introduce the selection mechanism Eq. 6, it can provably count at least $4$-paths and $4$-cycles.

**Proposition 3.3** (Counting paths and cycles). *Graph state space convolution Eq. 4 can at least count number of $3$-paths and $3$-cycles. With selection mechanism Eq. 6, it can at least count number of $4$-paths and $4$-cycles. Here "counting" means a node representation can express the number of paths starting at the node or the number of cycles involving the node.*

## 4 RELATED WORKS

**Linear Graph Transformers.** Graph transformers leverage attention mechanism (Vaswani et al., 2017; Dwivedi & Bresson, 2020; Kreuzer et al., 2021; Kim et al., 2022; Chen et al., 2022a) that can attend to all nodes in a graph, but yield quadratic complexity w.r.t. graph size. To reduce the complexity, Rampášek et al. (2022); Wu et al. (2022; 2023) adopt linear attention techniques, i.e., factorizing the attention kernel into products of Random Positive Features (Choromanski et al., 2020). Wu et al. (2024); Deng et al. (2024) replaces attention by inner products of learnable features, which do not leverage any positional encodings but use extra message passing layers to encode graph topology. For comparison, GSSC (Eq. 4) derived from SSMs is based on a factorization of convolution kernels, which shares some similar spirits. In fact, a recent work points out some equivalences between linear transformers and SSMs (Dao & Gu, 2024). However, GSSC is technically different from linear graph transformers. The latter uses random features to approximate the *specific* attention kernel (softmax+inner product), and generally the node features and positional encodings are both used in constructing the random features. In contrast, GSSC adopts positional encodings exclusively (may also includes node features if using selective mechanism Eq. 6) to construct convolution kernels in a learnable and stable way, and it is not restricted to the attention kernel. Finally, there are other works that use methods other than attention factorization: Shirzad et al. (2023) leverages virtual global nodes and expander graphs to perform sparse attention; Kong et al. (2023) applies a projection matrix to reduce the graph size factor $n$ to a lower dimension $k$. Both are very different from GSSC.

Table 1: Benchmark on GNN Benchmark & Long Range Graph Benchmark. **Bold**$^\dagger$, **Bold**$^\ddagger$, and **Bold** denote the first, second, and third best results, respectively. Results are reported as mean$_{\pm\text{std}}$.

| | MNIST | CIFAR10 | PATTERN | CLUSTER | MalNet-Tiny | PascalVOC-SP | Peptides-func | Peptides-struct |
|---|---|---|---|---|---|---|---|---|
| | Accuracy ↑ | Accuracy ↑ | Accuracy ↑ | Accuracy ↑ | Accuracy ↑ | F1 score↑ | AP ↑ | MAE↓ |
| GCN | $90.705_{\pm0.218}$ | $55.710_{\pm0.381}$ | $71.892_{\pm0.334}$ | $68.498_{\pm0.976}$ | $81.0$ | $0.1268_{\pm0.0060}$ | $0.5930_{\pm0.0023}$ | $0.3496_{\pm0.0013}$ |
| GIN | $96.485_{\pm0.252}$ | $55.255_{\pm1.527}$ | $85.387_{\pm0.136}$ | $64.716_{\pm1.553}$ | $88.98_{\pm0.56}$ | $0.1265_{\pm0.0076}$ | $0.5498_{\pm0.0079}$ | $0.3547_{\pm0.0045}$ |
| GAT | $95.535_{\pm0.205}$ | $64.223_{\pm0.455}$ | $78.271_{\pm0.186}$ | $70.587_{\pm0.447}$ | $92.10_{\pm0.24}$ | – | – | – |
| GatedGCN | $97.340_{\pm0.143}$ | $67.312_{\pm0.311}$ | $85.568_{\pm0.088}$ | $73.840_{\pm0.326}$ | $92.23_{\pm0.65}$ | $0.2873_{\pm0.0219}$ | $0.5864_{\pm0.0077}$ | $0.3420_{\pm0.0013}$ |
| SAN | – | – | $86.581_{\pm0.037}$ | $76.691_{\pm0.650}$ | | $0.3216_{\pm0.0027}$ | $0.6439_{\pm0.0075}$ | $0.2545_{\pm0.0012}$ |
| GraphGPS | $98.051_{\pm0.126}$ | $72.298_{\pm0.356}$ | $86.685_{\pm0.059}$ | $78.016_{\pm0.180}$ | $93.50_{\pm0.41}$ | $0.3748_{\pm0.0109}$ | $0.6535_{\pm0.0041}$ | $0.2500_{\pm0.0005}$ |
| Exphormer | $98.550^\dagger_{\pm0.039}$ | $74.690_{\pm0.125}$ | $86.740_{\pm0.015}$ | $78.070_{\pm0.037}$ | $94.02^\ddagger_{\pm0.21}$ | $0.3975_{\pm0.0037}$ | $0.6527_{\pm0.0043}$ | $0.2481_{\pm0.0007}$ |
| Grit | $98.108_{\pm0.111}$ | $76.468_{\pm0.881}$ | $87.196^\ddagger_{\pm0.076}$ | $80.026^\dagger_{\pm0.277}$ | – | – | $0.6988_{\pm0.0082}$ | $0.2460_{\pm0.0012}$ |
| GRED | $98.383_{\pm0.012}$ | $76.853^\ddagger_{\pm0.185}$ | $86.759_{\pm0.020}$ | $78.495^\ddagger_{\pm0.103}$ | – | – | $0.7133^\dagger_{\pm0.0011}$ | $0.2455^\ddagger_{\pm0.0013}$ |
| Graph-Mamba-I | $98.420_{\pm0.080}$ | $73.700_{\pm0.340}$ | $86.710_{\pm0.050}$ | $76.800_{\pm0.360}$ | $93.40_{\pm0.27}$ | $0.4191^\ddagger_{\pm0.0126}$ | $0.6739_{\pm0.0087}$ | $0.2478_{\pm0.0016}$ |
| GSSC | $98.492^\ddagger_{\pm0.051}$ | $77.642^\dagger_{\pm0.456}$ | $87.510^\dagger_{\pm0.082}$ | $79.156^\ddagger_{\pm0.152}$ | $94.06^\dagger_{\pm0.64}$ | $0.4561^\dagger_{\pm0.0039}$ | $0.7081^\ddagger_{\pm0.0062}$ | $0.2459^\ddagger_{\pm0.0020}$ |

**State Space Models (SSMs).** Classic SSMs (Kalman, 1960; Hyndman et al., 2008; Durbin & Koopman, 2012) describe the evolution of state variables over time using first-order differential equations or difference equations, providing a unified framework for time series modeling. Similar to RNNs (Pascanu, 2013; Graves, 2013; Sutskever et al., 2014), SSMs may also suffer from poor memorization of long contexts and long-range dependencies. To address this issue, Structural SSMs (S4) with a structural matrix $\bar{A}$ are introduced to capture long-range dependencies, e.g., HiPPO (Gu et al., 2020; 2021a) and diagonal matrices (Gupta et al., 2022; Gu et al., 2022). Many variants of SSMs (Gu & Dao, 2023; Mehta et al., 2022; Ma et al., 2022; Fu et al., 2022) are also proposed. See Wang et al. (2024b) for a comprehensive survey. Note that GSSC Eq. 4 does not reply on a structural matrix $\bar{A}$ to achieve long-range dependencies. This is because $\bar{A}$ serves to describe the casual (recurrence) relation Eq. 1 for sequences, while there is not such causality for graphs. Instead, GSSC is generalized from the SSM convolution Eq. 2 and its behavior relies on the design of convolution kernel, i.e., the inner product of learnable graph positional encodings. The long-range property can be achieved by choosing $\phi$ functions, as evidenced by Proposition 3.1.

**State Space Models for Graphs.** There are some efforts to replace the attention mechanism in graph transformers with SSMs. They mainly focus on tokenizing graphs and apply the existing SSM such as Mamba (Gu & Dao, 2023). Graph-Mamba-I (Wang et al., 2024a) sorts nodes into sequences by node degrees and applies Mamba. As node degrees could have multiplicity, this approach requires random permutation of sequences during training, and the resulting model is not permutation equivariant to node indices reordering. Graph-Mamba-II (Behrouz & Hashemi, 2024) extracts the $1, 2, ..., K$-hop subgraphs of a root node, treats each $k$-hop subgraph as a token. Each subgraph is assigned a representation using GNNs, and these subgraphs form a sequence for the root node. It then applies Mamba to this sequence to aggregate the representation of the root node and further applies Mamba to a sequence of root nodes to get graph representations, but the latter operation breaks permutation invariance. Ding et al. (2024) also treated $k$-hop neighbors with different $k$'s as a sequence while using Deepsets encoders (Zaheer et al., 2017) instead of GNNs. These approaches incur significant computational overhead as they require applying GNNs/DeepSets to encode every subgraph token first. Pan et al. (2024) focuses on heterogeneous graph scenarios, sorting and tokenizing rooted subgraphs based on the metapaths and applying Mamba to the sequentialized subgraphs. Zhao et al. (2024) aims at the design of graph spectral convolutions, applying SSMs to naturally ordered frequencies to build a graph filter. Compared to these methods, GSSC does not adopt any graph sequentialization but instead generalizes the causal state space convolution to graphs, preserving permutation equivariance and maintaining linear complexity.

## 5 EXPERIMENTS

We evaluate the effectiveness of GSSC on 13 datasets against various baselines. Particularly, we focus on answering the following questions:

- **Q1**: How expressive is GSSC in terms of counting graph substructures?
- **Q2**: How effectively does GSSC capture long-range dependencies?
- **Q3**: How does GSSC perform on general graph benchmarks compared to other baselines?
- **Q4**: How does the computational time/space of GSSC scale with graph size?

Below we briefly introduce the model implementation, included datasets and baselines, and a more detailed description can be found in Appendix B.

**Datasets.** To answer **Q1**, we use the graph substructure counting datasets from (Chen et al., 2020b; Zhao et al., 2021; Huang et al., 2022). Each of the synthetic datasets contains 5k graphs generated from different distributions (see Chen et al. (2020b) Appendix M.2.1), and the task is to predict the number of cycles as node-level regression. To answer **Q2**, we evaluate GSSC on Long Range Graph Benchmark (Dwivedi et al., 2022), which requires long-range interaction reasoning to achieve strong performance. Specifically, we adopt Peptides-func (graph-level classification with 10 functional labels of peptides), Peptides-struct (graph-level regression of 11 structural properties of molecules), and PascalVOC-SP (classify superpixels of image graphs into corresponding object classes). To answer **Q3**, molecular graph datasets (ZINC (Dwivedi et al., 2023) and ogbg-molhiv (Hu et al., 2020)), image graph datasets (MNIST, CIFAR10 (Dwivedi et al., 2023)), synthetic graph datasets (PATTERN, CLUSTER (Dwivedi et al., 2023)), and function call graphs (MalNet-Tiny) (Freitas et al., 2020), are used to evaluate the performance of GSSC. ZINC is a molecular property prediction (graph regression) task containing two partitions of dataset, ZINC-12K (12k samples) and ZINC-full (250k samples). Ogbg-molhiv consists of 41k molecular graphs for graph classification. CIFAR10 and MNIST are 8-nearest neighbor graph of superpixels constructed from images for classification. PATTERN and CLUSTER are synthetic graphs generated by the Stochastic Block Model (SBM) to perform node-level community classification. Finally, to answer **Q4**, we construct a synthetic dataset with graph sizes from 1k to 60k to evaluate how GSSC scales.

**GSSC Implementation.** We implement deep models consisting of GSSC blocks Eq. 4, where $\phi$ is DeepSets Zaheer et al. (2017). Each layer includes one MPNN (to incorporate edge features) and one GSSC block, followed by a nonlinear readout to merge the outputs of MPNN and GSSC. The resulting deep model can be seen as a GraphGPS (Rampášek et al., 2022) with the vanilla transformer replaced by GSSC. Selective mechanism is only introduced to cycle-counting tasks, because we find the GSSC w/o selective mechanism is already powerful and yields excellent results in real-world tasks. In our experiments, GSSC utilizes the smallest $d = 32$ eigenvalues and their eigenvectors for all datasets except molecular ones, which employ $d = 16$. See Appendix B for full details of model hyperparameters.

**Baselines.** We consider various baselines that can be mainly categorized into: (1) MPNNs: GCN (Kipf & Welling, 2016), GIN (Xu et al., 2018), GAT (Veličković et al., 2018), Gated GCN (Bresson & Laurent, 2017) and PNA (Corso et al., 2020); (2) Subgraph GNNs: NGNN (Zhang & Li, 2021), ID-GNN (You et al., 2021), GIN-AK+ (Zhao et al., 2021), I²-GNN (Huang et al., 2022); (3) Graph transformers: Graphormer (Ying et al., 2021), SAN (Kreuzer et al., 2021), GraphGPS (Rampášek et al., 2022), Exphormer (Shirzad et al., 2023), Grit (Ma et al., 2023); (4) Others: SUN (Frasca et al., 2022), Specformer (Bo et al., 2022), GRED (Ding et al., 2024), Graph-Mamba-I (Wang et al., 2024a), SignNet (Lim et al., 2022), SPE (Huang et al., 2024). Note that we also compare the baseline Graph-Mamba-II (Behrouz & Hashemi, 2024), and GSSC outperforms it on all included datasets, but we put their results in Appendix C as we cannot reproduce their results due to the lack of code.

### 5.1 GRAPH SUBSTRUCTURE COUNTING

Table 2 shows the normalized MAE results (MAE divided by the standard deviation of targets). We adopt GIN backbone for all baseline subgraph models (Huang et al., 2022). In terms of predicting 3-cycles and 4-cycles, GSSC achieves the best results compared to subgraph GNNs and I²-GNNs (models that can provably count 3, 4-cycles) validating Theorem 3.3. For the prediction of 5-cycles, GSSC greatly outperforms the MPNN and ID-GNN (models that cannot predict 5-cycles) reducing normalized MAE by 94.6% and 71.9%, respectively. Besides, GSSC achieves constantly better performance than GNNAK+, a subgraph

Table 2: Benchmark on graph substructure counting (normalized MAE ↓ ). Selective mechanism is applied for GSSC.

| | 3-Cycle | 4-Cycle | 5-Cycle |
|---|---|---|---|
| GIN | 0.3515 | 0.2742 | 0.2088 |
| ID-GIN | 0.0006 | 0.0022 | 0.0490 |
| NGNN | 0.0003 | 0.0013 | 0.0402 |
| GIN-AK+ | 0.0004 | 0.0041 | 0.0133 |
| I²-GNN | 0.0003 | 0.0016 | 0.0028 |
| Exphormer | 0.0006 | 0.0468 | 0.0827 |
| graph-Mamba-I | 0.0014 | 0.0113 | 0.0301 |
| GraphGPS (Transformer) | 0.0007 | 0.0125 | 0.0297 |
| graphGPS (Performer) | 0.0011 | 0.0131 | 0.0301 |
| GSSC | 0.0002 | 0.0013 | 0.0113 |

GNN model that is strictly stronger than ID-GNN and NGNN (Huang et al., 2022). These results demonstrate the empirically strong function-fitting ability of GSSC.

### 5.2 GRAPH LEARNING BENCHMARKS

Table 1 and Table 3 evaluate the performance of GSSC on multiple widely used graph learning benchmarks. GSSC achieves excellent performance on all benchmark datasets, and the result of each benchmark is discussed below.

**Long Range Graph Benchmark (Dwivedi et al., 2022).** We test the ability to model long-range interaction on PascalVOC-SP, Peptides-func, and Peptides-struct, as shown in Table 1. Remarkably, GSSC achieves *state-of-the-art* performance on PascalVOC-SP and delivers second-best results on the other two datasets, coming extremely close to the leading benchmarks. These results underscore GSSC's robust ability to capture long-range dependencies.

**Molecular Graph Benchmark (Dwivedi et al., 2023; Hu et al., 2020).** Table 3 shows the results on molecular graph datasets. GSSC achieves the best results on ZINC-Full and ogbg-molhiv and is comparable to the state-of-the-art model on ZINC-12k, which could be attributed to its great expressive power and stable positional encodings.

**GNN Benchmark (Dwivedi et al., 2023) & MalNet-Tiny (Freitas et al., 2020).** Table 1 (MNIST to CLUSTER) presents the results on the GNN Benchmark datasets and MalNet-Tiny. GSSC achieves *state-of-the-art* performance on CIFAR10, PATTERN, and MalNet-Tiny, and ranks second-best on MNIST and CLUSTER, demonstrating its superior capability on general graph-structured data.

**Ablation study.** The comparison to GraphGPS and MPNNs naturally serves as an ablation study: the proposed GSSC is replaced by a vanilla transformer or removed while other modules are identical. GSSC consistently outperforms MPNNs and GraphGPS on all tasks, validating the effectiveness of GSSC as an alternative to full attention in graph transformers.

Table 3: Benchmark on molecular datasets. **Bold**[†], **Bold**[‡], and **Bold** denote the first, second, and third best results, respectively.

| | ZINC-12k | ZINC-Full | ogbg-molhiv |
|---|---|---|---|
| | MAE ↓ | MAE ↓ | AUROC ↑ |
| GCN | $0.367_{\pm 0.011}$ | $0.113_{\pm 0.002}$ | $75.99_{\pm 1.19}$ |
| GIN | $0.526_{\pm 0.051}$ | $0.088_{\pm 0.002}$ | $77.07_{\pm 1.49}$ |
| GAT | $0.384_{\pm 0.007}$ | $0.111_{\pm 0.002}$ | – |
| PNA | $0.188_{\pm 0.004}$ | – | $79.05_{\pm 1.32}$ |
| NGNN | $0.111_{\pm 0.003}$ | $0.029_{\pm 0.001}$ | $78.34_{\pm 1.86}$ |
| GIN-AK+ | $0.080_{\pm 0.001}$ | – | $\mathbf{79.61}_{\pm 1.19}$ |
| I²-GNN | $0.083_{\pm 0.001}$ | $0.023_{\pm 0.001}$ | $78.68_{\pm 0.93}$ |
| SUN | $0.083_{\pm 0.003}$ | $0.024_{\pm 0.003}$ | $\mathbf{80.03}^{\ddagger}_{\pm 0.55}$ |
| Graphormer | $0.122_{\pm 0.006}$ | $0.052_{\pm 0.005}$ | – |
| SAN | $0.139_{\pm 0.006}$ | – | $77.85_{\pm 2.47}$ |
| GraphGPS | $0.070_{\pm 0.004}$ | – | $78.80_{\pm 1.01}$ |
| GraphGPS (Performer) | $0.072_{\pm 0.002}$ | – | $77.79_{\pm 1.25}$ |
| Exphormer | $0.111_{\pm 0.007}$ | – | $78.79_{\pm 1.31}$ |
| Specformer | $\mathbf{0.066}_{\pm 0.003}$ | – | $78.89_{\pm 1.24}$ |
| SPE | $0.070_{\pm 0.004}$ | – | – |
| SignNet | $0.084_{\pm 0.006}$ | $0.024_{\pm 0.003}$ | – |
| Grit | $\mathbf{0.059}^{\dagger}_{\pm 0.002}$ | $0.023_{\pm 0.001}$ | – |
| GRED | $0.077_{\pm 0.002}$ | – | – |
| Graph-Mamba-I | $\mathbf{0.067}_{\pm 0.002}$ | – | $78.23_{\pm 1.21}$ |
| GSSC | $\mathbf{0.064}^{\ddagger}_{\pm 0.002}$ | $\mathbf{0.019}^{\dagger}_{\pm 0.001}$ | $\mathbf{80.35}^{\dagger}_{\pm 1.42}$ |

### 5.3 COMPUTATIONAL COSTS COMPARISON

The computational costs of graph learning methods can be divided into two main components: 1) preprocessing, which includes operations such as calculating positional encoding, and 2) model training and inference. To demonstrate the efficiency of GSSC, we benchmark its computational costs for both components against 4 recent state-of-the-art methods, including GraphGPS (Rampášek et al., 2022), Grit (Ma et al., 2023), Exphormer (Shirzad et al., 2023), and Graph-Mamba-I (Wang et al., 2024a). Notably, Exphormer, Graph-Mamba-I, and GSSC are designed with linear complexity with respect to the number of nodes $n$, whereas Grit and GraphGPS exhibit quadratic complexity by design. According to our results below, GSSC is one of the most efficient models (even for large graphs) that can capture long-range dependencies. However, in practice, one must carefully consider whether the module that explicitly captures global dependencies is necessary for very large graphs.

**Benchmark Setup.** To accurately assess the scalability of the evaluated methods, we generate random graphs with node counts ranging from 1k to 60k. To simulate the typical sparsity of graph-structured data, we introduce $n^2 \times 1\%$ edges for graphs containing fewer than 10k nodes, and $n^2 \times 0.1\%$ edges for graphs with more than 10k nodes. For computations performed on GPUs, we utilize torch.utils.benchmark.Timer and

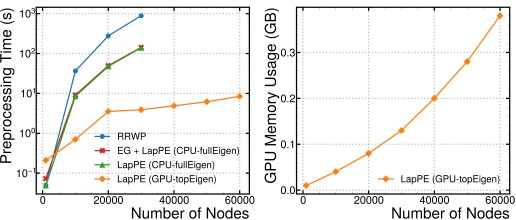

Figure 3: Preprocessing costs per graph.

torch.cuda.max_memory_allocated to measure time and space usage; for those performed on CPUs, time.time is employed. Results are averaged over more than 100 runs to ensure reliability. All methods are implemented using author-provided code and all experiments are conducted on a server equipped with Nvidia RTX 6000 Ada GPUs (48 GB) and AMD EPYC 7763 CPUs.

**Preprocessing Costs.** As all baseline methods implement preprocessing on CPUs, we first focus on CPU time usage to assess scalability, as illustrated in Fig. 3 (left). Grit computes RRWP (Ma et al., 2023), which notably requires more time due to its repeated multiplications between pairs of $n \times n$ matrices. Other methods, e.g., Exphormer, GraphGPS, Graph-Mamba-I, and GSSC, may use Laplacian eigendecomposition for graph positional encoding. For these, we follow the

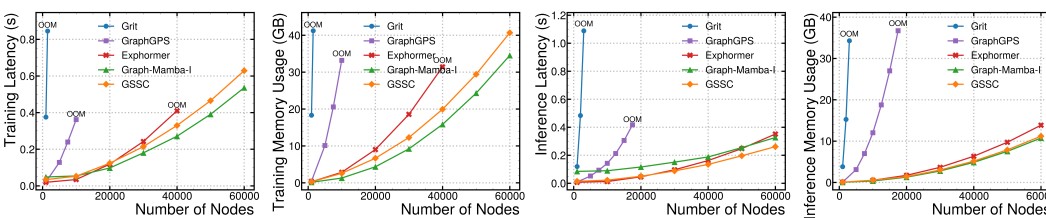

Figure 4: Model training and inference costs per graph.

implementation of baselines and perform full eigendecomposition on CPUs (limited to 16 cores), depicted by the green line in the figure. Exphormer additionally constructs expander graphs (EG), slightly increasing its preprocessing time (the red line). Clearly, full eigendecomposition is costly for large graphs; however, GSSC requires only the smallest $d$ eigenvalues and their eigenvectors. Thus, for large graphs, we can bypass full eigendecomposition and leverage iterative methods on GPUs, such as $\mathrm{torch.lobpcg}$ (Knyazev, 2001), to obtain the top-$d$ results, which is both fast and GPU memory-efficient (linear in $n$), shown by the orange lines (LapPE GPU-topEigen) in the figures with $d = 32$. Nonetheless, for small graphs, full eigendecomposition on CPUs remains efficient, aligning with practices in prior studies.

**Model Training/Inference Costs.** Fig. 4 benchmarks model training (forward + backward passes) and inference (forward pass only) costs in an inductive node classification setting. All methods are ensured to have the same number of layers and roughly 500k parameters. Methods marked "OOM" indicate out-of-memory errors on a GPU with 48 GB memory when the number of nodes further increases by 5k. GSSC and Graph-Mamba-I emerge as the two most efficient methods. Although Graph-Mamba-I shows slightly better performance during training, this advantage can be attributed to its direct integration with the highly optimized Mamba API (Gu & Dao, 2023), and GSSC's efficiency may also be further improved with hardware-aware optimization in the low-level implementation.

**Computational Costs of GSSC on Real-World Datasets.** The above benchmark evaluates graph (linear) transformers on very large graphs to thoroughly test their scalability. However, as in practice graph transformers are generally applied to smaller graphs (Rampášek et al., 2022), where capturing global dependencies can be more beneficial, here we also report the computational costs for two representative and widely used real-world benchmark datasets. The results are presented in Table 4, where ZINC-12k and

Table 4: Computational costs of GSSC on real-world datasets.

|  | ZINC-12k | PascalVOC-SP |
|---|---|---|
| Avg. # nodes | 23.2 | 479.4 |
| # graphs | 12,000 | 11,355 |
| # epochs | 2,000 | 300 |
| Training time per epoch | 10.9s | 13.9s |
| Total training time | 6.1h | 1.2h |
| Total preprocessing time | 20.6s | 334.8s |
| $\frac{\text{Total preprocessing time}}{\text{Total training time}}$ | 0.1% | 7.6% |

PascalVOC-SP are included as examples of real-world datasets with the smallest and largest graph sizes, respectively, for evaluating graph transformers. Preprocessing is done on CPUs per graph following previous works due to the relatively small graph sizes, and the number of training epochs used is also the same as prior studies (Rampášek et al., 2022; Ma et al., 2023; Shirzad et al., 2023). We find that the total preprocessing time is negligible for datasets with small graphs (e.g., ZINC-12k), comparable to the duration of a single training epoch (typically the number of training epochs is larger than 100, meaning a ratio $< 1\%$). For datasets with larger graphs (e.g., PascalVOC-SP), pre-processing remains reasonably efficient, consuming less than $10\%$ of the total training time. Notably, the preprocessing time could be further reduced by computing eigendecomposition on GPUs with batched graphs.

## 6 CONCLUSION AND LIMITATIONS

In this work, we study the extension of State Space Models (SSMs) to graphs. We propose Graph State Space Convolution (GSSC) that leverages global permutation-equivariant aggregation and factorizable graph kernels depending on relative graph distances. These operations naturally inherit the advantages of SSMs on sequential data: (1) efficient computation; (2) capability of capturing long-range dependencies; (3) good generalization for various sequence lengths (graph sizes). Numerical experiments demonstrate the superior performance and efficiency of GSSC.

One potential limitation of our work is that precisely computing full eigenvectors could be expensive for large graphs. See discussions in Sec. 3.2 and our empirical evaluation in Sec. 5.3 that shows good scalability even for large graphs with 60k nodes.

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

## A DEFERRED PROOFS

### A.1 PROOF OF PROPOSITION 3.1

**Proposition 3.1.** *There exists $\phi$ such that for GSSC Eq. 4, the gradient norm $\|\partial h_u/\partial x_v\|$ does not decay as $spd(u, v)$ grows, where $spd$ denotes shortest path distance.*

*Proof.* For simplicity, let us assume $m = 1$, i.e., hidden dimension is one, which makes $h_u, x_u$ scalars. Now let weights $\boldsymbol{W}_q = \boldsymbol{W}_k = \boldsymbol{W}_o = 1$ and $\boldsymbol{W}_{sq} = \boldsymbol{W}_{sk} = \boldsymbol{W}_s = 0$. Let $\phi(\lambda) = \sum_{k=1}^{n} c_k \lambda^k$, where $c_k > 0$ are arbitrary positive constants that do not decay to zero as $n, k \to \infty$. Then the derivative $\partial h_u/\partial x_v$ becomes

$$\frac{\partial h_u}{\partial x_v} = \frac{\partial}{\partial x_v} \sum_{k=1}^{n} A_{u,v}^k \sum_{v \in V} x_v = \sum_{k=1}^{n} c_k \#(\text{walks from } u \text{ to } v) \geq c_{spd(u,v)}. \tag{7}$$

Therefore, $\|\partial h_u/\partial x_v\| \geq |c_{spd(u,v)}|$. $\qquad\square$

### A.2 PROOF OF PROPOSITION 3.2

**Proposition 3.2.** *GSSC is strictly more powerful than WL test and not more powerful than 3-WL test.*

*Proof.* Let us show that GSSC is more powerful than WL test. Note that WL test has the same power as message passing GNNs (Xu et al., 2018), so it is sufficient to show that GSSC is more powerful than message passing GNNs. first, by letting $\phi(\boldsymbol{\Lambda}) = \Lambda$ and weights $\boldsymbol{W}_q = \boldsymbol{W}_k = I$, the convolution kernel $\langle z_u \boldsymbol{W}_q, z_v \boldsymbol{W}_k \rangle$ becomes $\boldsymbol{A}_{u,v}$, i.e., the entry of Adjacency matrix. Thus GSSC can mimic message passing GNNs. Besides, if we consider two nonisomorphic graphs: one consists of two triangles and one is a hexagon, clearly message passing GNNs cannot distinguish them while GSSC can with proper $\phi$ and weights. Therefore we claim that GSSC is more powerful than message passing GNNs and WL test as well.

Let us show that GSSC is not more powerful than 3-WL test. According to (Zhang et al., 2024), Corollary 4.5, any eigenspace projection GNNs, i.e., GNNs whose node-pair features are augmented by a basis invariant function of the PE of the node pairs, is not more powerful than 3-WL test. As GSSC leverages a basis invariant and stable function of PE as convolution kernel, it can be seen as an eigenspace projection GNN and thus is not more powerful than 3-WL. $\qquad\square$

### A.3 PROOF OF PROPOSITION 3.3

**Proposition 3.3** (Counting paths and cycles). *Graph state space convolution Eq. 4 can at least count number of 3-paths and 3-cycles. With selection mechanism Eq. equation 6, it can at least count number of 4-paths and 4-cycles. Here "counting" means the node representations can express the number of paths starting at the node or number of cycles involving the node.*

*Proof.* The number of cycles and paths can be expressed in terms of polynomials of adjacency matrix $\boldsymbol{A}$, as shown in (Perepechko & Voropaev, 2009). Specifically, let $[P_m]_{u,v}$ be number of length-$m$ paths starting at node $u$ and ending at node $v$, and $[C_m]_u$ be number of length-$m$ cycles involving node $u$, then

$$P_2 = \boldsymbol{A}^2, \tag{8}$$

$$C_3 = \text{diag}(\boldsymbol{A}^3), \tag{9}$$

$$P_3 = \boldsymbol{A}^3 + \boldsymbol{A} - \boldsymbol{A}\text{diag}(\boldsymbol{A}^2) - \text{diag}(\boldsymbol{A}^2)\boldsymbol{A}, \tag{10}$$

$$C_4 = \text{diag}(\boldsymbol{A}^4) + \text{diag}(\boldsymbol{A}^2) - \text{diag}(\boldsymbol{A}^2\text{diag}(\boldsymbol{A}^2)) - \text{diag}(\boldsymbol{A}\text{diag}(\boldsymbol{A}^2)\boldsymbol{A}), \tag{11}$$

$$P_4 = \boldsymbol{A}^4 + \boldsymbol{A}^2 + 3\boldsymbol{A} \odot \boldsymbol{A}^2 - \text{diag}(\boldsymbol{A}^3)\boldsymbol{A} - \text{diag}(\boldsymbol{A}^2)\boldsymbol{A}^2 - \boldsymbol{A}\text{diag}(\boldsymbol{A}^3) - \boldsymbol{A}^2\text{diag}(\boldsymbol{A}^2)$$
$$- \boldsymbol{A}\text{diag}(\boldsymbol{A}^2)\boldsymbol{A} \tag{12}$$

Here $\text{diag}(\cdot)$ means taking the diagonal of the matrix.

Now we are going to show that all these terms can be expressed by the GSSC kernel with specific choices of weights.

**2-paths and 3-cycles.** It is clear that GSSC Eq.4 can compute number of length-2 paths starting at node $u$, using $\sum_v [P_2]_{u,v} = \sum_v [\boldsymbol{A}^2]_{u,v} = \sum_v \langle \boldsymbol{\Lambda} \odot p_u, \boldsymbol{\Lambda} \odot p_v \rangle$. On other hand, number of 3-cycles $[C_3]_u = [\boldsymbol{A}^3]_{u,u} = \langle \boldsymbol{\Lambda}^{3/2} \odot p_u, \boldsymbol{\Lambda}^{3/2} p_u \rangle$ can also be implemented by Eq.equation 4 by setting $\boldsymbol{W}_s = 1$ and $\boldsymbol{W}_o = 0$.

**3-paths.** $\boldsymbol{A}^3$ and $\boldsymbol{A}$ can be expressed by the similar argument above. Note that $\sum_v [\boldsymbol{A}\mathrm{diag}(\boldsymbol{A}^2)]_{u,v} = \sum_v \boldsymbol{A}_{u,v}[\boldsymbol{A}^2]_{v,v}$. The term $[\boldsymbol{A}^2]_{v,v}$ can be represented by the node feature after one layer of GSSC, as argued in counting $[C_3]_u$. Therefore $\sum_v \boldsymbol{A}_{u,v}[\boldsymbol{A}^2]_{v,v} = \sum_v \langle \boldsymbol{\Lambda}^{1/2} \odot p_u, \boldsymbol{\Lambda}^{1/2} p_v \rangle [\boldsymbol{A}^2]_{v,v}$ can be expressed by a two-layer GSSC. Finally, the term $\sum_v [\mathrm{diag}(\boldsymbol{A}^2)\boldsymbol{A}]_{u,v} = [\boldsymbol{A}^2]_{u,u} \cdot \sum_v \langle \boldsymbol{\Lambda}^{1/2} \odot p_u, \boldsymbol{\Lambda}^{1/2} \odot p_v \rangle$ can be computed a one-layer GSSC, by first computing $[\boldsymbol{A}^2]_{u,u}$ and $\sum_v \langle \boldsymbol{\Lambda}^{1/2} \odot p_u, \boldsymbol{\Lambda}^{1/2} \odot p_v \rangle$ separately and use a intermediate nonlinear MLPs to multiply them.

**4-cycles.** The term $[\mathrm{diag}(\boldsymbol{A}^4)]_u = \boldsymbol{A}^4_{u,u} = \langle \boldsymbol{\Lambda}^2 p_u, \boldsymbol{\Lambda}^2 p_u \rangle$ can be implemented by one-layer GSSC by letting $\boldsymbol{W}_s = 1, \boldsymbol{W}_o = 0$. Same for $[\mathrm{diag}(\boldsymbol{A}^2)]_u$. The term $[\mathrm{diag}(\boldsymbol{A}^2\mathrm{diag}(\boldsymbol{A}^2))]_u = [\boldsymbol{A}^2]_{u,u}[\boldsymbol{A}^2]_{u,u}$ can be implemented by one-layer GSSC with a multiplication nonlinear operation. Finally, note that the last term

$$[\mathrm{diag}(\boldsymbol{A}\mathrm{diag}(\boldsymbol{A}^2)\boldsymbol{A})]_u = \sum_v \boldsymbol{A}_{u,v}\boldsymbol{A}^2_{v,v}\boldsymbol{A}_{v,u} = \sum_v \boldsymbol{A}_{u,v}\boldsymbol{A}_{u,v}\boldsymbol{A}^2_{v,v}. \tag{13}$$

We know that $x_v := \boldsymbol{A}^2_{v,v}$ can be encoded into node feature after one-layer GSSC. Now the whole term $\sum_v \boldsymbol{A}_{u,v}\boldsymbol{A}_{u,v}x_v$ can be transformed into

$$\begin{aligned}
\sum_v \boldsymbol{A}_{u,v}\boldsymbol{A}_{u,v}x_v &= \sum_v \langle \boldsymbol{\Lambda}^{1/2}p_u, \boldsymbol{\Lambda}^{1/2}p_v \rangle \langle \boldsymbol{\Lambda}^{1/2}p_u, \boldsymbol{\Lambda}^{1/2}p_v \rangle x_v \\
&= \langle \boldsymbol{\Lambda}^{1/2}p_u \sum_v, \langle \boldsymbol{\Lambda}^{1/2}p_u, \boldsymbol{\Lambda}^{1/2}p_v \rangle p_v \odot x_v \rangle = \langle \boldsymbol{\Lambda}^{1/2}p_u, \tilde{z}_u \rangle,
\end{aligned} \tag{14}$$

where $\tilde{z}_u$ is a node-feature-dependent PE, which can be implemented by Eq.6. The readout $\langle \boldsymbol{\Lambda}^{1/2}p_u, \tilde{z}_u \rangle$, again, can be implemented by letting $\boldsymbol{W}_o = 0$ and $\boldsymbol{W}_s = 1$.

**4-paths.** All terms can be expressed by the same argument in counting 3-paths, except $\boldsymbol{A} \odot \boldsymbol{A}^2$ and $\boldsymbol{A}\mathrm{diag}(\boldsymbol{A}^2)\boldsymbol{A}$. To compute $\sum_v [\boldsymbol{A} \odot \boldsymbol{A}^2]_{u,v} = \sum_v \boldsymbol{A}_{u,v}\boldsymbol{A}^2_{u,v}$, note that this follows the same argument as in Eq.14, with $x_v$ replaced by 1 and the second $\boldsymbol{A}_{u,v}$ replaced by $\boldsymbol{A}^2_{u,v}$. To compute $\sum_v [\boldsymbol{A}\mathrm{diag}(\boldsymbol{A}^2)\boldsymbol{A}]_{u,v}$, note that

$$\sum_v [\boldsymbol{A}\mathrm{diag}(\boldsymbol{A}^2)\boldsymbol{A}]_{u,v} = \sum_w \boldsymbol{A}_{u,w}\boldsymbol{A}^2_{w,w} \sum_v \boldsymbol{A}_{w,v}. \tag{15}$$

Therefore, we can first use one-layer GSSC to encode $\boldsymbol{A}^2_{w,w}$ and $\sum_v \boldsymbol{A}_{w,v}$ into node features, multiply them together to get $x_w = \boldsymbol{A}^2_{w,w} \cdot \sum_v \boldsymbol{A}_{w,v}$, and then apply another GSSC layer with kernel $\boldsymbol{A}_{u,w}$ to get desired output $\sum_w \boldsymbol{A}_{u,w}x_w$. □

## B EXPERIMENTAL DETAILS

### B.1 DATASETS DESCRIPTION

**Graph Substructure Counting (Chen et al., 2020b; Zhao et al., 2021; Huang et al., 2022)** is a synthetic dataset containing 5k graphs generated from different distributions (Erdős-Rényi random graphs, random regular graphs, etc. see (Chen et al., 2020b) Appendix M.2.1). Each node is labeled by the number of $3, 4, 5, 6$-cycles that involves the node. The task is to predict the number of cycles as node-level regression. The training/validation/test set is randomly split by 3:2:5.

**ZINC (Dwivedi et al., 2023)** (MIT License) has two versions of datasets with different splits. ZINC-subset contains 12k molecular graphs from the ZINC database of commercially available chemical compounds. These represent small molecules with the number of atoms between 9 and 37. Each node represents a heavy atom (28 atom types) and each edge represents a chemical bond (3 types). The

Table 5: Dataset statistics used in the experiments.

| Dataset | # Graphs | Avg. # nodes | Avg. # edges | Prediction level | Prediction task |
|---|---|---|---|---|---|
| Cycle-counting | 5,000 | 18.8 | 31.3 | node | regression |
| ZINC-subset | 12,000 | 23.2 | 24.9 | graph | regression |
| ZINC-full | 249,456 | 23.1 | 24.9 | graph | regression |
| ogbg-molhiv | 41,127 | 25.5 | 27.5 | graph | binary classif. |
| MNIST | 70,000 | 70.6 | 564.5 | graph | 10-class classif. |
| CIFAR10 | 60,000 | 117.6 | 941.1 | graph | 10-class classif. |
| PATTERN | 14,000 | 118.9 | 3,039.3 | node | binary classifi. |
| CLUSTER | 12,000 | 117.2 | 2,150.9 | node | 6-class classif. |
| MalNet-Tiny | 5,000 | 1,410.3 | 2,859.9 | graph | 5-class classif. |
| Peptides-func | 15,535 | 150.9 | 307.3 | graph | 10-class classif. |
| Peptides-struct | 15,535 | 150.9 | 307.3 | graph | regression |
| PascalVOC-SP | 11,355 | 479.4 | 2,710.5 | node | 21-class classif. |

task is to do graph-level regression on the constrained solubility (logP) of the molecule. The dataset comes with a predefined 10K/1K/1K train/validation/test split. ZINC-full is similar to ZINC-subset but with 250k molecular graphs instead.

**ogbg-molhiv (Hu et al., 2020)** (MIT License) are molecular property prediction datasets adopted by OGB from MoleculeNet (Wu et al., 2018). These datasets use a common node (atom) and edge (bond) featurization that represent chemophysical properties. The task is a binary graph-level classification of the molecule's fitness to inhibit HIV replication. The dataset split is predefined as in (Hu et al., 2020).

**MNIST and CIFAR10 (Dwivedi et al., 2023)** (CC BY-SA 3.0 and MIT License) are derived from image classification datasets, where each image graph is constructed by the 8 nearest-neighbor graph of SLIC superpixels for each image. The task is a 10-class graph-level classification and standard dataset splits follow the original image classification datasets, i.e., for MNIST 55K/5K/10K and for CIFAR10 45K/5K/10K train/validation/test graphs.

**PATTERN and CLUSTER (Dwivedi et al., 2023)** (MIT License) are synthetic datasets of community structures, sampled from the Stochastic Block Model. Both tasks are an inductive node-level classification. PATTERN is to detect nodes in a graph into one of 100 possible sub-graph patterns that are randomly generated with different SBM parameters than the rest of the graph. In CLUSTER, every graph is composed of 6 SBM-generated clusters, and there is a corresponding test node in each cluster containing a unique cluster ID. The task is to predict the cluster ID of these 6 test nodes.

**MalNet-Tiny (Freitas et al., 2020)** (CC-BY license) is a subset of the larger MalNet dataset, consisting of function call graphs extracted from Android APKs. It includes 5,000 graphs, each with up to 5,000 nodes, representing either benign software or four categories of malware. In this subset, all original node and edge attributes have been removed, and the goal is to classify the software type solely based on the graph structure.

**Peptides-func and Peptides-struct (Dwivedi et al., 2022)** (MIT License) are derived from 15k peptides retrieved from SATPdb (Singh et al., 2016). Both datasets use the same set of graphs but the prediction tasks are different. Peptides-func is a graph-level classification task with 10 functional labels associated with peptide functions. Peptides-struct is a graph-level regression task to predict 11 structural properties of the molecules.

**PascalVOC-SP (Dwivedi et al., 2022)** (MIT License) is a node classification dataset based on the Pascal VOC 2011 image dataset (Everingham et al., 2010). Superpixel nodes are extracted using the SLIC algorithm (Achanta et al., 2012) and a rag-boundary graph that interconnects these nodes are constructed. The task is to classify the node into corresponding object classes, which is analogous to the semantic segmentation.

## B.2 Random Seeds and Dataset Splits

All included datasets have standard training/validation/test splits. We follow previous works reporting the test results according to the best validation performance, and the results of every dataset are evaluated and averaged over five different random seeds (Rampášek et al., 2022; Ma et al., 2023; Shirzad et al., 2023). Due to the extremely long running time of ZINC-Full (which requires over 80 hours to train one seed on an Nvidia RTX 6000 Ada since it uses 2k epochs for training following previous works (Rampášek et al., 2022; Ma et al., 2023)), its results are averaged over three random seeds.

## B.3 Hyperparameters

Table 6, 7, 8, and 9 detail the hyperparameters used for experiments in Sec. 5. We generally follow configurations from prior works (Rampášek et al., 2022; Shirzad et al., 2023). Notably, the selective mechanism (i.e., Eq. 6) is only employed for graph substructure counting tasks, and GSSC utilizes the smallest $d = 32$ eigenvalues and their eigenvectors for all datasets except molecular ones, which use $d = 16$. Consistent with previous research (Rampášek et al., 2022; Ma et al., 2023), we also maintain the number of model parameters at around 500k for the ZINC, PATTERN, CLUSTER, and LRGB datasets, and approximately 100k for the MNIST and CIFAR10 datasets.

Since our implementation is based on the framework of GraphGPS (Rampášek et al., 2022), which combines the learned node representations from the MPNN and the global module (GSSC in our case) in each layer, dropout is applied for regularization to the outputs from both modules, as indicated by MPNN-dropout and GSSC-dropout in the hyperparameter tables.

Table 6: Model hyperparameters for graph substructure counting datasets.

| Hyperparameter | 3-Cycle | 4-Cycle | 5-Cycle |
|---|---|---|---|
| # Layers | 4 | 4 | 4 |
| Hidden dim | 96 | 96 | 96 |
| MPNN | GatedGCN | GatedGCN | GatedGCN |
| Lap dim $d$ | 16 | 16 | 16 |
| Selective | True | True | True |
| Batch size | 256 | 256 | 256 |
| Learning Rate | 0.001 | 0.001 | 0.001 |
| Weight decay | 1e-5 | 1e-5 | 1e-5 |
| MPNN-dropout | 0.3 | 0.3 | 0.3 |
| GSSC-dropout | 0.3 | 0.3 | 0.3 |
| # Parameters | 926k | 926k | 926k |

Table 7: Model hyperparameters for molecular property prediction datasets.

| Hyperparameter | ZINC-12k | ZINC-Full | ogbg-molhiv |
|---|---|---|---|
| # Layers | 10 | 10 | 6 |
| Hidden dim | 64 | 64 | 64 |
| MPNN | GINE | GINE | GatedGCN |
| Lap dim $d$ | 16 | 16 | 16 |
| Selective | False | False | False |
| Batch size | 32 | 128 | 32 |
| Learning Rate | 0.001 | 0.002 | 0.002 |
| Weight decay | 1e-5 | 0.001 | 0.001 |
| MPNN-dropout | 0 | 0.1 | 0.3 |
| GSSC-dropout | 0.6 | 0 | 0 |
| # Parameters | 436k | 436k | 351k |

Table 8: Model hyperparameters for datasets from Long Range Graph Benchmark (LRGB) (Dwivedi et al., 2022).

| Hyperparameter | PascalVOC-SP | Peptides-func | Peptides-struct |
|---|---|---|---|
| # Layers | 4 | 4 | 4 |
| Hidden dim | 96 | 96 | 96 |
| MPNN | GatedGCN | GatedGCN | GatedGCN |
| Lap dim $d$ | 32 | 32 | 32 |
| Selective | False | False | False |
| Batch size | 32 | 128 | 128 |
| Learning Rate | 0.002 | 0.003 | 0.001 |
| Weight decay | 0.1 | 0.1 | 0.1 |
| MPNN-dropout | 0 | 0.1 | 0.1 |
| GSSC-dropout | 0.5 | 0.1 | 0.3 |
| # Parameters | 375k | 410k | 410k |

Table 9: Model hyperparameters for datasets from GNN Benchmark (Dwivedi et al., 2023) and MalNet-Tiny (Freitas et al., 2020).

| Hyperparameter | MNIST | CIFAR10 | PATTERN | CLUSTER | MalNet-Tiny |
|---|---|---|---|---|---|
| # Layers | 3 | 3 | 24 | 24 | 5 |
| Hidden dim | 52 | 52 | 36 | 36 | 64 |
| MPNN | GatedGCN | GatedGCN | GatedGCN | GatedGCN | GatedGCN |
| Lap dim $d$ | 32 | 32 | 32 | 32 | 32 |
| Selective | False | False | False | False | False |
| Batch size | 16 | 16 | 32 | 16 | 16 |
| Learning Rate | 0.005 | 0.005 | 0.001 | 0.001 | 0.0015 |
| Weight decay | 0.01 | 0.01 | 0.1 | 0.1 | 0.001 |
| MPNN-dropout | 0.1 | 0.1 | 0.1 | 0.3 | 0.1 |
| GSSC-dropout | 0.1 | 0.1 | 0.5 | 0.3 | 0.3 |
| # Parameters | 133k | 131k | 539k | 539k | 299k |

Table 10: Comparing with previous graph mamba works. **Bold**[†] denotes the best results. Results are reported as mean$_{\pm\text{std}}$.

| | ZINC-12k | MNIST | CIFAR10 | PATTERN | CLUSTER | PascalVOC-SP | Peptides-func | Peptides-struct |
|---|---|---|---|---|---|---|---|---|
| | MAE $\downarrow$ | Accuracy $\uparrow$ | Accuracy $\uparrow$ | Accuracy $\uparrow$ | Accuracy $\uparrow$ | F1 score $\uparrow$ | AP $\uparrow$ | MAE $\downarrow$ |
| Graph-Mamba-I | NaN | $98.420_{\pm 0.080}$ | $73.700_{\pm 0.340}$ | $86.710_{\pm 0.050}$ | $76.800_{\pm 0.360}$ | $0.4191_{\pm 0.0126}$ | $0.6739_{\pm 0.0087}$ | $0.2478_{\pm 0.0016}$ |
| Graph-Mamba-II | N/A | $98.390_{\pm 0.180}$ | $75.760_{\pm 0.420}$ | $87.140_{\pm 0.120}$ | N/A | $0.4393_{\pm 0.0112}$ | $0.7071_{\pm 0.0083}$ | $0.2473_{\pm 0.0025}$ |
| GSSC | $\mathbf{0.064}^{\dagger}_{\pm 0.002}$ | $\mathbf{98.492}^{\dagger}_{\pm 0.051}$ | $\mathbf{77.642}^{\dagger}_{\pm 0.456}$ | $\mathbf{87.510}^{\dagger}_{\pm 0.082}$ | $\mathbf{79.156}^{\dagger}_{\pm 0.152}$ | $\mathbf{0.4561}^{\dagger}_{\pm 0.0039}$ | $\mathbf{0.7081}^{\dagger}_{\pm 0.0062}$ | $\mathbf{0.2459}^{\dagger}_{\pm 0.0020}$ |

# C SUPPLEMENTARY EXPERIMENTS

In this section, we present supplementary experiments comparing GSSC with previous graph mamba works, i.e., Graph-Mamba-I (Wang et al., 2024a) and Graph-Mamba-II (**?**). As shown in Table 10, GSSC significantly outperforms both models on all datasets. We attempted to evaluate these works on additional datasets included in our experiments, such as ZINC-12k, but encountered challenges. Specifically, Graph-Mamba-II has only an empty GitHub repository available, and Graph-Mamba-I raises persistent NaN (Not a Number) errors when evaluated on other datasets. Our investigation suggests that these errors of Graph-Mamba-I stem from fundamental issues in their architecture and implementation, and there is no easy way to fix them, i.e., they are not caused by any easy-to-find risky operations such as $\log(\text{small negative numbers})$ or $\frac{1}{0}$. Consequently, Graph-Mamba-I (and Graph-Mamba-II) may potentially exhibit severe numerical instability and cannot be applied to some datasets. Prior to encountering the NaN error, the best observed MAE for Graph-Mamba-I on ZINC-12k was $\sim 0.10$.

