# OpenReview forum: "What Can We Learn from State Space Models for Machine Learning on Graphs?"
_ICLR.cc/2025/Conference — Submitted to ICLR 2025_

### Official Review · Reviewer_CutC · 2024-10-30

**Soundness:** 3
**Presentation:** 3
**Contribution:** 3
**Rating:** 6
**Confidence:** 3

**Summary:**

This paper proposes Graph State Space Convolution (GSSC), an extension of state-space models to graph-structured data. The authors emphasize GSSC’s capability for capturing long-range dependencies in linear time and demonstrate competitive performance across several benchmarks.

**Strengths:**

1. The paper achieves competitive performance compared to SOTA methods.
2. The paper proposes a novel approach in extending state-space models to graphs.
3. The paper provides a plausible framework for capturing long-range dependencies efficiently.

**Weaknesses:**

My main concerns relate to the complexity claims, specifically:
1. **Layer Complexity vs Expressivity:** The paper states that the complexity of a GSSC layer is “… $O(nmd)$    where $n$ is the number of nodes and $m$, $d$ are hidden and positional encoding dimension.” (239-240) This means that a GSSC layer has $O(|V|)$ complexity   . Consequently, GSSC is in general *incapable of examining every edge in the graph*, unlike MPNNs with $O(|V|+|E|)$ complexity, such as GINE [1]. Although the authors prove that GSSC is “more powerful than MPNNs” *in terms of the WL hierarchy*, the expressivity implications of not being able to examine every edge seems to be overlooked. Even preprocessing the graph to incorporate edge features into nodes, which itself requires $O(|E|)$ time, would still necessitate $m\in O(\frac{|E|}{|V|})$ to store these features without information loss, thus exceeding $O(|V|)$ complexity overall.
2. **Preprocessing Complexity:** The paper claims that finding the top $d$ eigenpairs with Lanczos methods has $O(nd^2)$ complexity (286-287). However, since sparse matrix-vector multiplication with the Laplacian matrix is necessary for Lanczos methods, the complexity per iteration would be at least $O(|E|)$, resulting in an overall complexity of at least $O(d|E|)$ to find $d$ eigenpairs. This exceeds the paper’s claim of $O(nd^2)$ preprocessing, and thus requires clarification.

Despite these concerns, the paper’s main contributions remain valid. I would appreciate if the authors could clarify these points during the rebuttal phase.

[1] https://arxiv.org/abs/1905.12265

**Questions:**

Please see weaknesses.

---

> ### Author Response · Authors · 2024-11-22
> **Author Response**
>
> We thank the reviewer for the insightful comments. Here is our response.
>
> > **Weakness 1**: Layer Complexity vs Expressivity: The paper states that the complexity of a GSSC layer is “… O(nmd)    where n is the number of nodes and m, d are hidden and positional encoding dimension.” (239-240) This means that a GSSC layer has O(|V|) complexity. Consequently, GSSC is in general incapable of examining every edge in the graph, unlike MPNNs with O(|V|+|E|) complexity, such as GINE. Although the authors prove that GSSC is “more powerful than MPNNs” in terms of the WL hierarchy, the expressivity implications of not being able to examine every edge seems to be overlooked. Even preprocessing the graph to incorporate edge features into nodes, which itself requires O(|E|) time, would still necessitate m∈O(|E||V|) to store these features without information loss, thus exceeding O(|V|) complexity overall.
>
> Note that GSSC is *practically* linear-time as we set d (number of eigenvectors) to be a constant in actual implementation. In *theory*,  GSSC with constant d many eigenvectors can also distinguish non-isomorphic graphs that 1-WL cannot distinguish. It is true that in our proof, n many eigenvectors are required as a sufficient condition to losslessly simulate MPNNs, which means the complexity of eigendecomposition preprocessing is O(nE). But practically it is unnecessary, especially given that we also incorporate GSSC with MPNNs in our actual implementation.
>
> > **Weakness 2**: Preprocessing Complexity: The paper claims that finding the top d eigenpairs with Lanczos methods has O(nd2) complexity (286-287). However, since sparse matrix-vector multiplication with the Laplacian matrix is necessary for Lanczos methods, the complexity per iteration would be at least O(|E|), resulting in an overall complexity of at least O(d|E|) to find d eigenpairs. This exceeds the paper’s claim of O(nd2) preprocessing, and thus requires clarification.
>
> Thank you for pointing this out. We made a typo and the complexity of Lanczos methods is indeed O(Ed), which is still considered efficient for sparse graphs. We clarify that in the revised manuscript.

---

> > ### Comment · Reviewer_CutC · 2024-11-24
> >
> > I would like to thank the authors for their response.
> >
> > > Note that GSSC is practically linear-time as we set d (number of eigenvectors) to be a constant in actual implementation.
> >
> > While I trust the authors that a constant value of $d$ is sufficient for the presented experiments, the relationship between $d$ and model performance remains unclear, as the paper does not present *ablation studies* on different values of $d$ on graphs with different sizes or densities. The paper also provides no *theoretical account* on why $d$ could be kept constant. Therefore, I am not fully convinced that $d$ could be set constant on very large graphs (\>1M nodes, such as [ogbn-products](https://ogb.stanford.edu/docs/nodeprop/)), or graphs with higher densities (denser than the %0.1~%1 used in experiments of sec. 5.3). I acknowledge that there might be insufficient time for new experiments, but I hope that the authors could provide a convincing argument of why $d$ could be kept constant.
> >
> > > We made a typo and the complexity of Lanczos methods is indeed O(Ed), which is still considered efficient for sparse graphs. We clarify that in the revised manuscript.
> >
> > Could the authors explain why `lobpcg` has $O(n)$ space complexity (L498-500)? Is storing the sparse adjacency matrix, which is $O(|E|)$, not required?

---

> > > ### Author Response · Authors · 2024-11-26
> > > **Response to Official Comment by Reviewer CutC**
> > >
> > > We would like to sincerely thank the reviewer for reading our response and providing new feedbacks.
> > > >  I hope that the authors could provide a convincing argument of why $d$ could be kept constant.
> > >
> > > There are some justifications that constant $d$ is still helpful:
> > > - the top $d$ smallest eigenvalues, even for super large graphs, provide richer information than MPNNs cannot capture. For example, taking the top $d$ smallest eigenvalues allows to determine if the number of connected components is larger than $d$, which cannot be expressed/computed by MPNNs. Therefore, practically when we incorporate GSSC with MPNNs in practice, a constant number of eigenvalues/eigenvectors still boosts the expressive power of plain MPNNs.
> > > - There are also empirical evidence on real-world graphs that many physical quantities are graph signals with low frequencies [1] and GNNs can achieve the best performance using only low frequencies [2].
> > >
> > > > Could the authors explain why lobpcg has $O(n)$ space complexity (L498-500)? Is storing the sparse adjacency matrix, which is $O(|E|)$, not required?
> > >
> > > Theoretically, logbpcg can be implemented in a matrix-free manner, which means it does not need to store the adjacency matrix $A$, but instead accesses $A$ by evaluating matrix-vector products $x^{\top}Ax$. Therefore, the space complexity is simply storing the eigenvectors $x$ in memory, which has space complexity $O(n)$. Practically, we can store the matrix $A$ directly into memory if it is not too large, then the space complexity could become $O(|E|)$.
> > >
> > >
> > > [1] Sandryhaila, A., & Moura, J. M. (2014). Discrete signal processing on graphs: Frequency analysis. IEEE Transactions on signal processing, 62(12), 3042-3054.
> > >
> > > [2] Nt, H., & Maehara, T. (2019). Revisiting graph neural networks: All we have is low-pass filters. arXiv preprint arXiv:1905.09550.

---

> > > > ### Comment · Reviewer_CutC · 2024-11-26
> > > >
> > > > I would like to thank the authors for responding. I have some follow-up questions:
> > > >
> > > > > Theoretically, logbpcg can be implemented in a matrix-free manner, which means it does not need to store the adjacency matrix $A$ , but instead accesses $A$ by evaluating matrix-vector products . Therefore, the space complexity is simply storing the eigenvectors $x$ in memory, which has space complexity $O(n)$.
> > > >
> > > > Does this claim agree with figure 4? I’m not sure that the relationship is linear, as claimed in the paper, because the slope of the orange line increases as $n$ increases. In addition, does `torch.lobpcg` (L498 of your paper) work with a matrix *not* stored in memory? I don’t see how that is possible in the [code](https://github.com/pytorch/pytorch/blob/42ab61241edd244051d8e4a1a9f8ccfaa095caca/torch/_lobpcg.py). Finally, the eigenvectors of which matrix does GSSC require: the adjacency matrix $A$ or the Laplacian matrix $L$?

---

> ### Author Response · Authors · 2024-11-26
> **Further Response to Official Comment by Reviewer CutC**
>
> Thank you for the follow-up questions.
>
> > the slope of the orange line increases as $n$ increases. Does this claim agree with figure 4?
>
> Figure 4 shows the training time/inference time, which is not related to preprocessing (eigendecomposition). Note that this evaluation is on random graphs with $E=n^2\cdot0.0001$. It does not seem linear to $n$ because GSSC incorporates MPNN layers in practice, the latter of which grows linear w.r.t. number of edges. The GSSC block itself is linear w.r.t. $n$, unlike $O(n^2)$ for graph transformers.
>
>
> > does torch.lobpcg (L498 of your paper) work with a matrix not stored in memory?
>
> In principle, we can implement matrix-free by definining the matrix to be a linear operator object (i.e., a object which can take a input vector and return a output vector). Packages like scipy.sparse.linalg.lobpcg can support this. Torch.lobpcg does not support this function for now (https://github.com/pytorch/pytorch/issues/89255). In our implementation we just stored the matrix into memory as the graphs are not large.
>
> > which matrix does GSSC require: the adjacency matrix $A$ or the Laplacian matrix $L$?
>
> The normalized Laplacian $L$. It has the similar level of sparsity as $A$.

---

> ### Comment · Reviewer_CutC · 2024-11-26
>
> Thank the authors for their response.
>
> Sorry for the misunderstanding, I meant figure 3, not figure 4. The paper claims (L498-500):
> >…to obtain the top-dresults, which is both fast and
> GPU memory-efficient (linear in n), shown by the orange lines (LapPE GPU-topEigen) in the figures with $d=32$.
>
> If it’s referring to figure 3 (# nodes vs GPU memory usage, preprocessing), it still does not look linear as claimed. The slope of the orange line increases as # nodes increases.
>
> > The normalized Laplacian $L$. It has the similar level of sparsity as $A$.
>
> > Theoretically, logbpcg can be implemented in a matrix-free manner, which means it does not need to store the adjacency matrix $A$ , but instead accesses $A$ by evaluating matrix-vector products . Therefore, the space complexity is simply storing the eigenvectors $x$ in memory, which has space complexity $O(n)$.
>
> In that case, is $A$ meant to be $L$ in your response above? It does not change the analysis but I’m slightly confused by the inconsistency with the paper (the paper decomposes $L$ while $A$ is considered in your response).

---

> > ### Author Response · Authors · 2024-11-27
> > **Further Response to Official Comment by Reviewer CutC**
> >
> > Thank you for your comments.
> >
> > > If it’s referring to figure 3 (# nodes vs GPU memory usage, preprocessing), it still does not look linear as claimed. The slope of the orange line increases as # nodes increases.
> >
> > We think it is not strictly linear to $n$ because we directly store Laplacian $L$ in GPU, which causes a linear space complexity w.r.t. $|E|$. This can be avoided by applying a matrix-free stype implementation as we discussed.
> >
> > > is $A$ meant to be $L$ in your response above?
> >
> > Yes, sorry for the confusion. I said $A$ because the logbpcg API uses notation $A$ for a sparse matrix, which actually means $L$ in our case.

---

> ### Comment · Reviewer_CutC · 2024-11-28
>
> Thank the authors for their response.
>
> I can better understand the complexities of relevant algorithms now, but this reflects major issues in the clarity of the paper. Here are some issues that came up in our conversation:
> 1. The paper incorrectly claims $O(nd^2)$ time complexity for preprocessing (286-287), because the complexity of Lanczos iteration was misrepresented (fixed in revision).
> 2. The paper claims $O(n)$ space complexity for preprocessing (L498-500), but the implementation has a space complexity of $O(|E|)$. This is not explicitly discussed in the paper.
> 3. To support the above claim, the paper presents Figure 3, and claims that memory consumption is linear to $n$. Meanwhile, the orange line of Figure 3 is clearly not linear, due to the actual implementation requiring $O(|E|)$ memory.
>
> These inconsistencies would likely mislead a casual reader, and prompt a more serious reader to question the complexity claims of the paper. Therefore, I strongly recommend that the authors address these issues, and preferably, check all complexity claims in the paper for similar issues.

---

> > ### Author Response · Authors · 2024-11-30
> >
> > Thank you very much for your constructive suggestions regarding the complexity argument in this work. We will incorporate your feedback and revise the manuscript accordingly in its next version. Many thanks again!

---

> ### Comment · Reviewer_CutC · 2024-12-02
>
> My overall rating still reflects my evaluation of the paper in its current form. However, I have increased my soundness score to acknowledge the authors’ efforts in clarifying their claims.

---

### Official Review · Reviewer_wW9i · 2024-11-02

**Soundness:** 3
**Presentation:** 2
**Contribution:** 2
**Rating:** 3
**Confidence:** 4

**Summary:**

Machine learning on graphs often uses Message Passing Neural Networks (MPNNs), but these have limited expressive power. Graph State Space Convolution (GSSC) is proposed as a new method that extends State Space Models (SSMs) to graph data, overcoming these limitations and demonstrating superior performance on benchmark datasets. GSSC achieves good results on many datasets and offers a scalable solution for graph machine learning.

**Strengths:**

1. **Novelty**: Unlike previous methods that design SSM variants on graphs, this paper approaches the problem from the perspective of separable distance kernels on graphs. This idea is both novel and interesting.

2. **Comprehensive Evaluations**: The experimental section of this paper covers synthetic datasets, real datasets, and computational efficiency benchmarks, providing a comprehensive evaluation.

**Weaknesses:**

1. **Experimental Design**: My main concern is that the model evaluated in the experiments is a hybrid of GSSC and MPNN, without showing the performance of GSSC alone. This makes it difficult to assess the independent contribution of GSSC, affecting the judgment of its effectiveness. I strongly recommend that the authors provide experimental results for the GSSC module alone, which will help readers understand the primary contributions of GSSC.

2. **Baseline Selection**: In Section 5.1, the authors compare only with MPNN, without providing baseline comparisons with SSM or GT, which seems insufficient. Considering the authors claim that GSSC is a replacement module for Transformers, I suggest adding comparisons with other models that include shortest path information (such as high-order MPNN) or global information (such as Graph Transformer). Moreover, the current baselines are mainly single MPNN models, and comparing them with the hybrid model (GSSC+MPNN) seems unfair.

3. **Inconsistency in Model Architecture**: In Section 5.1, the authors use selective GSSC, but it is not used in other benchmarks.If different architectural variants of the model are used in the experiments, I advise clearly indicating this in the tables. Additionally, the authors claim that GSSC without selective is already powerful enough, does this imply that using selective GSSC would be better? If so, it is recommended to provide relevant experimental results to support this argument. If selective GSSC is not used due to other disadvantages (such as computational efficiency), it would be beneficial to discuss this further in the paper. Generally, a unified model architecture is more attractive than one that requires adjustments across different benchmarks.

**Questions:**

1. Compared to other graph SSM works, this study indeed offers a novel perspective. In your opinion, what advantages does this new viewpoint provide over other methods? How do your experiments support these advantages?

2. The title is "What Can We Learn from State Space Models for Machine Learning on Graphs." Could you elaborate a bit more on other graph SSM models / SSM model on other domins, and compare them with GSSC?

---

> ### Author Response · Authors · 2024-11-22
> **Author Response**
>
> We thank the reviewer for the valuable comments. Here is our response.
>
> > **Weakness 1**: Experimental Design: My main concern is that the model evaluated in the experiments is a hybrid of GSSC and MPNN, without showing the performance of GSSC alone.
>
> Thank you for the suggestion. Note that the general goal of SSM (and the proposed GSSC) is to replace vanilla attention, which does not have a good interface to incorporate edge (token-pairwise) features. This is an inherent limitation of SSMs. To address this limitation, SSMs (and their graph extension GSSC) need to be paired with modules that can handle token-pairwise features. Indeed, most of the graph transformer [1,2] and graph SSMs [3, 4] baselines are also built upon a hybrid with MPNNs to handle edge features.
>
>
>
> > **Weakness 2**: Baseline Selection: In Section 5.1, the authors compare only with MPNN, without providing baseline comparisons with SSM or GT, which seems insufficient.
>
> Thank you for the constructive suggestion. Previously we have comparisons to graph Mamba and graph transformers in Section 5.2, Tables 1 and 3. Now we also add graph transformer (GraphGPS), sparse/linear graph transformer (Exformer, GraphGPS+performer) and graph SSMs (Graph-mamba-I) in Section 5.1, counting experiments. Please see *Table A and Table B in the General Response*. Note that High-order MPNNs like I2-GNN were already included in Section 5.1 in the original manuscript.
> Notably, we find that though these new baselines have relatively low training error, their test error is many times larger than GSSC. This is because these models do not preserve permutation equivariance, causing a poor generalization ability. In contrast, GSSC properly handles positional encodings in a permutation equivariant and stable manner, and thus has better generalization.
>
> > **Weakness 3**: Inconsistency in Model Architecture: In Section 5.1, the authors use selective GSSC, but it is not used in other benchmarks. If different architectural variants of the model are used in the experiments, I advise clearly indicating this in the tables.
>
> Thank you for the suggestion. We now highlight in the Table caption if the selective mechanism is applied. For the counting task, we applied the selective mechanism because it is crucial for a higher expressive power. For real-world tasks, we found the selective mechanism does not bring significant gain and we guess expressive power is not a main bottleneck there. For model consistency, we will include the results of selective mechanisms on real-world benchmarks in the final version of the manuscript.
>
> > **Question 1**: Compared to other graph SSM works, this study indeed offers a novel perspective. In your opinion, what advantages does this new viewpoint provide over other methods? How do your experiments support these advantages?
>
> Note that we have already discussed the difference between GSSC and other graph SSMs in Section 4, Line 350. Compared to other graph SSMs, the biggest difference is that we do not tokenize the graph and apply existing SSMs for sequences. Instead, we provide a way to design SSMs specific for graph data. As a result, our method inherently preserves permutation equivariance, an important inductive bias for graphs, while previous graph SSMs fail to do so. This advantage leads to a better generalization, supported by results of our experiments.
>
> > **Question 2**: The title is "What Can We Learn from State Space Models for Machine Learning on Graphs." Could you elaborate a bit more on other graph SSM models / SSM model on other domins, and compare them with GSSC?
>
> We have a comprehensive comparison to other graph SSMs in Related Works, Line 350.
>
>
>
> [1] Rampášek, L., Galkin, M., Dwivedi, V. P., Luu, A. T., Wolf, G., & Beaini, D. (2022). Recipe for a general, powerful, scalable graph transformer. Advances in Neural Information Processing Systems, 35, 14501-14515.
>
> [2] Chen, D., O’Bray, L., & Borgwardt, K. (2022, June). Structure-aware transformer for graph representation learning. In International Conference on Machine Learning (pp. 3469-3489). PMLR.
>
> [3] Wang, C., Tsepa, O., Ma, J., & Wang, B. (2024). Graph-mamba: Towards long-range graph sequence modeling with selective state spaces. arXiv preprint arXiv:2402.00789.\
>
> [4] Behrouz, A., & Hashemi, F. (2024, August). Graph mamba: Towards learning on graphs with state space models. In Proceedings of the 30th ACM SIGKDD Conference on Knowledge Discovery and Data Mining (pp. 119-130).

---

> > ### Comment · Reviewer_wW9i · 2024-11-26
> > **Official Comment by Reviewer wW9i**
> >
> > Thanks for the response and revisions to the manuscript. However, my following concerns remain unaddressed:
> >
> > **Weakness 1.** I still believe it is necessary to showcase the performance of the GSSC as a single model. While GraphGPS [1] and EXPHORMER [2] are also hybrid models, they have demonstrated the performance of their proposed methods as single models. Not to mention, GRED [3], as a single model without using edge features, also exhibits strong performance in Table 1. Even if many datasets come with edge features, the GSSC single model can still be compared with single models that do not utilize edge features. Such an apples-to-apples comparison would better help readers understand the contribution of GSSC and the robustness of its model architecture.
> >
> > **Weakness 2.** Similar to the above, I also hope to see the performance of the GSSC single model in Table 2, so that it can be more fairly compared with a range of MPNN and high-order MPNN baselines.
> >
> > **Weakness 3.** The authors have committed to showcasing the performance of selective GSSC in Table 1, and I look forward to seeing the reported results.
> >
> > Therefore, I prefer to maintain my score.
> >
> > ---
> >
> > [1] Recipe for a General, Powerful, Scalable Graph Transformer, NeurIPS 2022
> >
> > [2] Exphormer: Sparse Transformers for Graphs, ICML 2023
> >
> > [3] Recurrent distance filtering for graph representation learning, ICML 2024

---

### Official Review · Reviewer_N4o9 · 2024-11-05

**Soundness:** 2
**Presentation:** 3
**Contribution:** 2
**Rating:** 5
**Confidence:** 4

**Summary:**

The paper introduces the Graph State Space Convolution (GSSC) method, an extension of State Space Models (SSMs) to graph data. GSSC utilizes global permutation-equivariant set aggregation and factorizable graph kernels based on relative node distances as its convolution kernels.

**Strengths:**

I have found the paper well written and self-contained. I think a non-expert could find most of the information in the paper, and I appreciate this aspect.

 Figures 1 and 2, which demonstrate the problem domain and architecture, are interesting and easy to read. I commend the authors for their explicit effort in making these illustrations clear and informative.

The insights are didactical and well communicated. The conclusion given by the experiments looks interesting and valuable for future practitioners, while I think a synthesis would be beneficial for the reader.

**Weaknesses:**

Despite these merits, I have the following concerns about the paper.

1- While there is a careful analysis of the different design decisions/performance tradeoffs, I feel that there is only a limited understanding about what are the properties of the Architecture that lead to these decisions/performance differences.

2-  Scalability Concerns: The paper acknowledges the challenge of scalability for larger graphs. To address this, the authors could explore methods for optimizing the computational complexity of the GSSC architecture.

3- Weak experimental study:  The paper lacks experimental evaluation of the Graph State Space Convolution (GSSC) method on heterophilic datasets. This omission is significant as such studies are crucial to assess GSSC's performance with heterophilic data and its robustness against issues like over-squashing and over-smoothing, which are common challenges in graph data analysis.

**Questions:**

(i) For the scalable version of the architecture when you proposed using the first k eigenvectors, how does the method address the issue of missing eigenvectors in the middle and end of the spectrum?

(ii) What are the main challenges in extending GSSC to heterophilic graphs, and do you have any preliminary insights on how these challenges could be addressed?

---

> ### Author Response · Authors · 2024-11-22
> **Author Response**
>
> We thank the reviewer for the valuable suggestions. Here is our response.
>
> > **Weakness 1**: While there is a careful analysis of the different design decisions/performance tradeoffs, I feel that there is only a limited understanding about what are the properties of the Architecture that lead to these decisions/performance differences.
>
> The motivation to design the GSSC architecture is by generalizing the key components (absolute positions, factorizable kernels and global sum) in SSMs to graph data, as we argued in Section 3.1, Line 211 to 215. Such design leads to many theoretical advantages, as shown in Remark 3.1, that: (1) GSSC is permutation equivariant, while most linear graph transformers and graph SSMs are not; (2) GSSC is stable to graph perturbation, leading to a better generalization ability; (3) GSSC can capture long-range dependencies (Proposition 3.1).  We believe these merits contribute to the performance gain.
>
> > **Weakness 2**: Scalability Concerns: The paper acknowledges the challenge of scalability for larger graphs. To address this, the authors could explore methods for optimizing the computational complexity of the GSSC architecture.
>
> The main computation bottleneck is from preprocessing of eigendecompositions. Ways to alleviate the costs of eigendecompositions are discussed in Section 3.2, Line 282, such as using top K eigenvectors with advanced eigenvalue decomposition algorithms.  Comparisons of actual preprocessing time was shown in Figure 3. Notably, our empirical evaluation shows that the practical preprocessing time is relatively small or negligible compared to total training time (Section 5.3, Line 509).
>
>
> > **Weakness3**: Weak experimental study: The paper lacks experimental evaluation of the Graph State Space Convolution (GSSC) method on heterophilic datasets. This omission is significant as such studies are crucial to assess GSSC's performance with heterophilic data and its robustness against issues like over-squashing and over-smoothing, which are common challenges in graph data analysis.
>
> We respectfully **disagree** with the reviewer on the “Weak experimental study”. Though we agree that heterophilic data (as well as over-squashing/over-smoothing problems) is important, it is not widely considered/adopted by our relevant works/baselines [1,2,3,4]. On the other hand, our work studies generalizing SSMs to graph data, with a focus on achieving efficiency and long-range dependencies on graph data. To demonstrate these claimed benefits, we have extensive experiments covering substructure counting (to examine expressive power and verify Proposition 3.3), long-range benchmark (to verify Proposition 3.1), well-accepted graph benchmarks such as ZINC, OGB-MOLHIV, and runtime evaluation. How to deal with heterophilic data has unique challenges and is out of the scope of our main focus.
>
> > **Question 1**: For the scalable version of the architecture when you proposed using the first k eigenvectors, how does the method address the issue of missing eigenvectors in the middle and end of the spectrum?
>
> Top-K eigenvectors is a well-adopted technique in positional-encodings-based methods. One justification is that many graph distances, e.g., diffusion distance, biharmonic distance, are dominated by low-frequency eigenvectors [5]. In other words, graph distances can be well approximated by top K eigenvalues and eigenvectors.
>
>
> > **Question 2**: What are the main challenges in extending GSSC to heterophilic graphs, and do you have any preliminary insights on how these challenges could be addressed?
>
> We believe that heterophilic graphs are another challenging scenario with many unique problems to consider, which are out of scope of our work.
>
>
> [1] Shirzad, H., Velingker, A., Venkatachalam, B., Sutherland, D. J., & Sinop, A. K. (2023, July). Exphormer: Sparse transformers for graphs. In International Conference on Machine Learning (pp. 31613-31632). PMLR.
>
> [2] Wang, C., Tsepa, O., Ma, J., & Wang, B. (2024). Graph-mamba: Towards long-range graph sequence modeling with selective state spaces. arXiv preprint arXiv:2402.00789.
>
> [3] Rampášek, L., Galkin, M., Dwivedi, V. P., Luu, A. T., Wolf, G., & Beaini, D. (2022). Recipe for a general, powerful, scalable graph transformer. Advances in Neural Information Processing Systems, 35, 14501-14515.
>
> [4] Ma, L., Lin, C., Lim, D., Romero-Soriano, A., Dokania, P. K., Coates, M., ... & Lim, S. N. (2023, July). Graph inductive biases in transformers without message passing. In International Conference on Machine Learning (pp. 23321-23337). PMLR.
>
> [5] Kreuzer, D., Beaini, D., Hamilton, W., Létourneau, V., & Tossou, P. (2021). Rethinking graph transformers with spectral attention. Advances in Neural Information Processing Systems, 34, 21618-21629.

---

### Official Review · Reviewer_TB1r · 2024-11-11

**Soundness:** 2
**Presentation:** 2
**Contribution:** 2
**Rating:** 5
**Confidence:** 4

**Summary:**

This work extends the state space models (SSMs) from sequence modeling to the domain of  graph-structured data. By tailoring SSMs for graphs, the proposed model (GSSC)
can capture long-range dependencies and overcome the limitations of Message Passing Neural Networks (MPNNs) while offering efficient computation addressing the quadratic computation of the graph transformers. To preserve the permutation equivariance in the graphs, it outlines a method for designing a permutation-invariant kernel for convolution operations on graphs. Furthermore, it extends to a data-dependent version of the proposed model by defining a selection mechanism for graph-structured data. The proposed model demonstrates provably stronger expressiveness than MPNNs and Graph Spectral Convolution in counting graph substructures.

**Strengths:**

- The paper provide a method to customize the recursion in the SSMs for graphs,
- It proposes a method for designing a permutation-invariant kernel for efficient global convolution operations on graphs.
- The paper demonstrates provably stronger expressiveness of the proposed model for 3-paths and 3-cycles and also for 4-paths and 4-cycles.

**Weaknesses:**

1. The distinction between the proposed model (equation 4) and linear graph attention is minimal. It is known, even prior to Dao & Gu (2024), that SSMs can be represented as linear attention. The global convolution representation used here is also an equivalent form of SSMs. Additionally, the use of positional encodings is not novel, as it is a feature used by other works such as GraphGPS, which propose a general framework for transformer-based models and can adopt their approximations like Performer and linear attention.

2. The presentation requires improvement, and some claims need to be more precise. For example:
   - a. at line 143, the *Computational efficiency and parallelism*,  The text should clarify that SSMs like S4, when represented as global convolution (equation 2), they can leverage the FFT algorithm to result in parallel and efficient quasi-linear complexity. Parallel scan is used for the recurrence form of equation (1) under certain conditions (employed by data-dependent SSMs like Mamba), which also results in O(n log n) computation but offers parallelization.
   - b. The statement that the kernel should be factorizable as a dot product to be permutation-invariant (line 127),  needs revision. The dot product between absolute position representations is one way to achieve translation invariance. Reference [2] offers alternative methods, like cross-correlation, to achieve translation invariance in kernels for global convolution using translation-equivariant functions.
   - c. The factorized form in line 259 is not equivalent to the data-dependent convolution presented in the preceding line.
   - d. The new positional encodings in equation (6) are positional encodings of all nodes but utilize only the features of node u (as it is a function of $x_u$ only). Lines 267 and 267 need correction to reflect this.
   - e.The text should elaborate on how $\phi()$ in  in eqn (5)  is modeled to be a permutation equivariant function and capture interactions between frequencies.




3. **Insufficient Empirical Studies**: The literature review lacks citations for many state-of-the-art models, and the experimental section lacks comparisons to them.
   - Since the proposed model is compared with GSC and Linear Graph Transformers in the paper, it is essential to include their performance comparison in the experimental section (particularly in Table 2 for graph substructure counting, where the proposed model is expected to show superior expressiveness).
   - Comparisons with newer models like Spatial-Spectral GNN [1] and Polynormer [3] are also necessary.
   - Additionally, comparisons against other SSM-based models (Graph-Mamba I and II), GSC, Linear Graph Transformers and recent models in Tables 2 and 3 are important to demonstrate the proposed model empirically.

**Questions:**

1. Please address the aforementioned concerns
2. Some typos:
   - Equation 5 (line 201): the first term doesn’t require parentheses.
   - “reply on” -> rely on in line 268 and 346.
   - There are duplicate references for Behrouz & Hashemi, 2024


Conclusion:

While the proposed idea is appealing, and I acknowledge its potential impact, I am not sure that the paper is ready for ICLR in its current form. Therefore, I am hesitant to fully support acceptance but I am willing to increase my score if the concerns and questions are addressed.

**References:**
[1] Geisler, S., Kosmala, A., Herbst, D., & Günnemann, S. (2024). Spatio-Spectral Graph Neural Networks. arXiv preprint arXiv:2405.19121.
[2] M. Karami, A. Ghodsi, Orchid: Flexible and Data-Dependent Convolution for Sequence Modeling .” In Thirty-eighth Conference on Advances in Neural Information Processing Systems 2024.
[3] Deng, Chenhui, Zichao Yue, and Zhiru Zhang. "Polynormer: Polynomial-Expressive Graph Transformer in Linear Time." The Twelfth International Conference on Learning Representations.

Update 1: added references

---

> ### Author Response · Authors · 2024-11-22
> **Author Response**
>
> We thank the reviewer for the insightful comments. Here is our response.
>
> > **Weakness 1**: The distinction between the proposed model (equation 4) and linear graph attention is minimal.
>
> First, though conceptually related, GSSC is technically different from existing linear graph  attention. Linear attention aims to approximate the softmax attention kernel by random features. Particularly, the linear attention cannot handle the symmetry of positional encodings properly as they are treated as regular node features, which breaks permutation equivariance. In contrast, GSSC uses flexible and learnable positional encodings as the features in a permutation equivariant and stable manner, and they are not tied to specific kernels.
>
> Finally, to see the **empirical difference** between these architectures, we add extra comparisons to graph linear attention (e.g., GraphGPS+performer, Exphormer) on substructures counting and molecular graph benchmark. Please kindly see Tables A and B in our General Response. GraphGPS+performer has constantly worse performance than GraphGPS, as it is an approximation of the latter. In contrast, GSSC has a consistently superior performance over GraphGPS, GraphGPS+performer and Exphormer.
>
> >  **Weakness 2**: The presentation requires improvement, and some claims need to be more precise.
>
> We thank the reviewer for the detailed comment. For bullet point (a), we add discussions on the FFT algorithm in the revised manuscript. For bullet point (b), it seems the reviewer did not provide the mentioned reference [2] about cross correlation. We would like to include it if the reviewer can provide the reference. For bullet points (c-d), we find these are two typos and we now fix them. For bullet point (e), we use Deepsets as the phi function to model interaction between eigenvalues and we will elaborate it in the main text, Experiment Section.
>
> > **Weakness 3.1** : Since the proposed model is compared with GSC and Linear Graph Transformers in the paper, it is essential to include their performance comparison in the experimental section (particularly in Table 2 for graph substructure counting, where the proposed model is expected to show superior expressiveness).
>
> Thank you for the suggestion. We add extra comparisons to graph linear attention (e.g., GraphGPS+performer, Exphormer) on substructure counting and molecular graph benchmark.  Please kindly see Tables A and B in our General Respons above.
>
> > **Weakness 3.2**: Comparisons with newer models like Spatial-Spectral GNN and Polynormer are also necessary.’
>
> Thank you for introducing the models. Spatio-Spectral GNN [1] also has results on peptides-func and peptides-struct, which we will include in Table 1 in the next version of the manuscript. However, since it is not open-sourced, we would not be able to do further comparison given the limited time of rebuttal. Polynormer [3] has no overlapping datasets with us and we are working on adapting its code in our pipeline. If time allows we will update the results before the end of rebuttal.
>
> > **Weakness 3.3**: Additionally, comparisons against other SSM-based models (Graph-Mamba I and II), GSC, Linear Graph Transformers and recent models in Tables 2 and 3 are important to demonstrate the proposed model empirically.
>
> Thank you for your suggestion. We add extra comparisons to graph linear attention (e.g., GraphGPS+performer, Exphormer) and graph SSMs (Graph-Mamba I) on substructure counting and molecular graph benchmark. Please see Table A and Table B in our General Response.
>
> > **Question2**: about typos.
>
> Thank you for pointing them out.  The parentheses in Equation 5 (line 201) is to emphasize how the efficient computation is achieved. We fix the other two typos in the revised manuscript.
>
>
> [1] Geisler, S., Kosmala, A., Herbst, D., & Günnemann, S. (2024). Spatio-Spectral Graph Neural Networks. arXiv preprint arXiv:2405.19121.
>
> [3] Deng, C., Yue, Z., & Zhang, Z. (2024). Polynormer: Polynomial-expressive graph transformer in linear time. arXiv preprint arXiv:2403.01232.

---

> > ### Comment · Reviewer_TB1r · 2024-11-26
> >
> > I would like to thank the authors for their detailed response. I just wanted to clarify on the following points:
> >
> > > Linear Transformer (attention):
> >
> > By "Linear Transformer," I am referring to the family of alternative layers  to Transformers that are recently known as Linear Transformers, not those that scale linearly such as Performer (although this was also called Linear Transformer in [6] due to its linear scalability).
> > These layers replace the $exp(q^T k)$ with the dot-products $ϕ(q)^Tϕ(k)$, where the feature map can be a simple identity function (Linear Kernel) [1, 5], $elu(x) + 1$ [2], a sigmoid function in Polynormer [3], or a Taylor expansion [4].
> >  Given that these linear forms can be reformulated as recurrent neural networks (RNNs) or State Space Models (SSMs) [1], the proposed SSM-based model in equation (4) can be considered a specific instance within this category of models and seems very similar to the Linear Kernel in [1, 5].
> >
> > > Question 2 (d): Parentheses in Equation (5)
> >
> > The current placement of parentheses in equation (5) implies that the summation is computed solely over $z_v W_k$. However, I believe the summation should include the entire expression $\\sum\_{v}z\_v W\_k \\odot W\_o x\_v$.
> >
> > Furthermore, my original review is revised to include the references.
> >
> > I appreciate your attention to these matters and update the manuscript accordingly.
> >
> >
> > **References:**
> > [1] Yang, S., Wang, B., Shen, Y., Panda, R. and Kim, Y., Gated Linear Attention Transformers with Hardware-Efficient Training. In Forty-first International Conference on Machine Learning.
> > [2] Katharopoulos, A., Vyas, A., Pappas, N. and Fleuret, F., 2020, November. Transformers are rnns: Fast autoregressive transformers with linear attention. In International conference on machine learning
> > [3] Deng, Chenhui, Zichao Yue, and Zhiru Zhang. "Polynormer: Polynomial-Expressive Graph Transformer in Linear Time." The Twelfth International Conference on Learning Representations.
> > [4] Arora, S., Eyuboglu, S., Zhang, M., Timalsina, A., Alberti, S., Zou, J., Rudra, A. and Re, C., Simple linear attention language models balance the recall-throughput tradeoff. In Forty-first International Conference on Machine Learning.
> > [5] Yutao Sun, Li Dong, Shaohan Huang, Shuming Ma, Yuqing Xia, Jilong Xue, Jianyong Wang, and Furu Wei. Retentive network: A successor to transformer for large language models

---

> > > ### Author Response · Authors · 2024-11-26
> > > **Further Author Response**
> > >
> > > We sincerely appreciate Reviewer TB1r for reading our rebuttal, making comments and introducing extra references. We have additionally cited these papers in the revised version. We would like to further response to the points mentioned above:
> > >
> > > > Linear Transformer (attention):
> > >
> > > - We agree that linear attentions (replace attention by inner products of feature maps, e.g., [1, 2, 3]) and SSMs can be reformulated to each others, and both of them adopt the idea of kernel factorization. GSSC, generalized from SSMs for sequences, shares some similar spirits of kernel factorization.
> > > - However, GSSC aims to generalizes SSMs to **graphs** and it proposes to do the factorization by leveraging a careful design of graph **stable positional encodings** that are specific to graphs. This generalization is inspired by our key observations of SSMs's usage of absolute positions (Line211). As a result, GSSC is technically different from the methods that directly apply linear attention ideas to graph transformers, e.g., [4] (which cannot handle graph positional encodings). In fact, the linear kernel [1,3] or $elu(x)+1$ [2] cannot be naively applied to positional encodings, as they break permutation equivariance and cause instability [5].
> > >
> > > > Question 2 (d): Parentheses in Equation (5)
> > >
> > > Thank you for correcting this point and it is indeed a typo. We remove the parentheses in the revised manuscript.
> > >
> > >
> > > [1] Yang, S., Wang, B., Shen, Y., Panda, R. and Kim, Y., Gated Linear Attention Transformers with Hardware-Efficient Training. In Forty-first International Conference on Machine Learning.
> > >
> > > [2] Katharopoulos, A., Vyas, A., Pappas, N. and Fleuret, F., 2020, November. Transformers are rnns: Fast autoregressive transformers with linear attention. In International conference on machine learning
> > >
> > > [3]  Yutao Sun, Li Dong, Shaohan Huang, Shuming Ma, Yuqing Xia, Jilong Xue, Jianyong Wang, and Furu Wei. Retentive network: A successor to transformer for large language models
> > >
> > > [4] Deng, Chenhui, Zichao Yue, and Zhiru Zhang. "Polynormer: Polynomial-Expressive Graph Transformer in Linear Time." The Twelfth International Conference on Learning Representations.
> > >
> > > [5] Wang, H., Yin, H., Zhang, M., & Li, P. (2022). Equivariant and stable positional encoding for more powerful graph neural networks. arXiv preprint arXiv:2203.00199.

---

### Author Response · Authors · 2024-11-22
**General Response**

We would like to express our sincere thanks to the reviewers for their constructive feedbacks.

Although graph transformers and graph SSMs, as the most relevant baselines, are included in our experiments, there is a concern regarding the lack of comparisons with graph linear transformers. To adress these concerns, here we provide some extra comparisons to graph linear transformers. Specifically, we consider GraphGPS+Performer, which means the transformer module is replaced by Performer. Note that Graph-mamba-II is not open-sourced so we do not include it here. The new results and other revisions during rebuttal are indicated correspondingly in the revised manuscript with blue text.

For substructure counting, we find that though these new baselines have relatively low training error, their test error is many times larger than GSSC. This is because these models do not preserve permutation equivariance, causing a poor generalization ability. In contrast, GSSC properly handles positional encodings in a permutation equivariant and stable manner, and thus have better generalization. For molecular graphs, GSSC is consistently better than all other graph linear transformers and graph SSMs. Note that similar to what we have discussed in Appendix C, Graph-Mamba-I suffers from significant numerical issues, as also questioned by others in their GitHub issue (https://github.com/bowang-lab/Graph-Mamba/issues/6). We observe that for all datasets it may converge to NaN randomly. In our experiments, for 5 out of 10 seeds, it may converge to NaN. Therefore, to collect Graph-Mamba-I’s results, we restart the training with a different seed if for a seed it eventually converges to NaN.


**Table A**: Subtructure Counting.
| Model |3-Cycle |4-Cycle |5-Cycle|
| --------  | -------- | -------- | -------- |
Exphormer | $0.0006$|$0.0468$|$0.0827$|
Graph-Mamba-I  | $0.0014$|$0.0113$|$0.0301$|
GraphGPS (Transformer) |$0.0007$|$0.0125$|$0.0297$|
GraphGPS (Performer) | $0.0011$|$0.0131$|$0.0301$|
GSSC (ours) |$0.0002$|$0.0013$|$0.0113$|

**Table B**: Benchmark on molecular datasets.
| Model |ZINC |OGBG-MOLHIV|
| --------  | -------- | -------- |
Exphormer | $0.111\pm0.007$|$78.79\pm1.31$|
Graph-Mamba-I  | $0.067\pm0.002$|$78.23\pm1.21$|
GraphGPS (Transformer) |$0.070\pm 0.004$|$78.80\pm1.01$|
GraphGPS (Performer) | $0.072\pm0.002$|$77.79\pm 1.25$|
GSSC (ours) |$0.064\pm 0.002$|$80.35\pm1.42$|

---

### Meta-Review · Area_Chair_3KXv · 2024-12-22

**Metareview:**

The paper does not receive consistently positive support from the reviewers even though after extensive discussions.

**Additional Comments On Reviewer Discussion:**

Most of the major issues remain unsolved. One of the reviewers was willing to raise the score slightly after discussions while other reviewers are not convinced.

---

### Decision · Program_Chairs · 2025-01-22

Reject